

# Retrieval of the raindrop size distribution from polarimetric radar data using double-moment normalisation

Timothy H. Raupach and Alexis Berne

Environmental Remote Sensing Laboratory, School of Architecture, Civil, and Environmental Engineering, École Polytechnique Fédérale de Lausanne (EPFL), 1015 Lausanne, Switzerland

*Correspondence to:* Alexis Berne (alexis.berne@epfl.ch)

**Abstract.** A new technique for estimating the raindrop size distribution (DSD) from polarimetric radar data is proposed. Two statistical moments of the DSD are estimated from polarimetric variables, and the DSD is reconstructed. The technique takes advantage of the relative invariance of the double-moment normalised DSD. The method was tested using X-band radar data and networks of disdrometers in three different climatic regions. Radar-derived estimates of the DSD compare reasonably well to observations. In the three tested domains, the proposed method performs similarly to and often better than a state-of-the-art DSD-retrieval technique. The approach is flexible because no specific double-normalised DSD model is prescribed. In addition, a method is proposed to treat noisy radar data to improve DSD-retrieval performance with radar measurements.

## 1 Introduction

The raindrop size distribution (DSD) describes the microstructure of liquid precipitation, and is highly variable (Jameson and Kostinski, 2001; Uijlenhoet et al., 2003; Jaffrain and Berne, 2012). The DSD is measured at the point scale by disdrometers. For applications such as numerical weather prediction (e.g. Baldauf et al., 2011) or radar remote sensing (e.g. Bringi and Chandrasekar, 2001) it is often necessary to know the areal DSD at the pixel scale. In other cases, such as studies of the microphysics of precipitation (Pruppacher and Klett, 2000), it would be useful to be able to remotely infer the DSD aloft or in remote locations. For these reasons, retrieval of the DSD from radar data has been a long-standing goal. In this paper we present a new technique for DSD retrieval from polarimetric radar data, which is based on the double-moment normalisation technique of Lee et al. (2004).

Polarimetric weather radars are particularly useful for remote retrieval of the DSD, because differences between vertically and horizontally polarised electromagnetic waves reflected off hydrometeors in the atmosphere provide information on the particles' concentration, size, and shape. In rainfall, radar reflectivity in horizontal ($Z_H$ [dBZ]) or vertical ($Z_V$ [dBZ]) polarisation primarily relates to drop concentration and size. Differential reflectivity ($Z_{DR}$ [dB]) reflects drop shape, and specific differential phase shift ($K_{dp}$ [° km$^{-1}$]) relates to both the concentration and shape of the drops (Bringi and Chandrasekar, 2001). Seliga and Bringi (1976) showed that $Z_{DR}$ can be linked to the median volume drop diameter, a microphysical property of rain. Since then, many methods for DSD retrieval from radar variables have been proposed.





Zhang et al. (2001) introduced the "constrained gamma" method, in which the shape and slope parameters of a gamma DSD model (Ulbrich, 1983) are assumed dependent. This assumption is subject to debate (e.g. Zhang et al., 2003; Atlas and Ulbrich, 2006; Moisseev and Chandrasekar, 2007; Cao and Zhang, 2009). The technique, modified by Brandes et al. (2003), can provide useful DSD information (Brandes et al., 2004a). In the "beta" method (Gorgucci et al., 2002), the effective slope of the drop

axis ratio to diameter relationship is retrieved. The slope is used to find parameter values for the normalised gamma model of Willis (1984), which has advantages for use with polarimetric observations (Illingworth and Blackman, 2002). Retrieval of the gamma model shape parameter with the beta method is subject to high uncertainty (Gorgucci et al., 2002; Anagnostou et al., 2008). To deal with noisy $Z_{DR}$ and $K_{dp}$ data at low rain rates, Bringi et al. (2002, 2003) used the beta method for heavy rain and disdrometer-based regressions on $Z_H$ and $Z_{DR}$ for light rain. Brandes et al. (2004b) found that the constrained gamma

method was in better agreement with disdrometer data than the beta method, while Anagnostou et al. (2008) reported similar performance from the two techniques, and both studies noted that the beta method is sensitive to errors in $K_{dp}$. Vulpiani et al. (2006) developed a neural-network DSD-retrieval technique, and spatial correlations of DSD model parameters have been retrieved from radar data (Thurai et al., 2012; Bringi et al., 2015).

X-band polarimetric weather radars are popular due to their portability, small size, and high resolution and sensitivity, but

measurements at X-band suffer from attenuation by heavy rain (Kalogiros et al., 2013; Anagnostou et al., 2013) and must be corrected (Matrosov et al., 2005; Park et al., 2005a). Several DSD-retrieval algorithms have been developed for X-band (e.g. Park et al., 2005b; Gorgucci et al., 2008; Kalogiros et al., 2013; Anagnostou et al., 2013), including some with integrated attenuation correction (e.g. Testud et al., 2000; Yoshikawa et al., 2014). The self-consistent with optional parameterization attenuation correction and microphysics estimation (SCOP-ME) algorithm, developed through studies by Anagnostou et al.

(2009, 2010) and Kalogiros et al. (2013), uses relationships calculated for the Rayleigh limit, corrected for Mie scattering at X-band. It performs well compared to contemporary algorithms and disdrometer observations (Anagnostou et al., 2013). In this paper we present a new method for DSD retrieval that uses the double-moment DSD normalisation of Lee et al. (2004), and compare it to SCOP-ME.

The rest of this manuscript is organised as follows: we briefly describe the double-moment DSD normalisation technique

of Lee et al. (2004) in Section 2. Bulk rainfall variables that we use are introduced in Section 3. Data used are presented in Section 4. In Section 5 we propose a new DSD-retrieval method that uses double-moment normalisation to retrieve the DSD from polarimetric radar data. Its performance is compared to that of SCOP-ME using radar variables simulated from DSD measurements in Section 6. In Section 7 we introduce a new method to reduce the effects of noise in radar measurements. Using this method, the DSD-retrieval algorithms are compared using radar data in Section 8. Conclusions are made in Section

30 9.

## 2 Double-moment DSD normalisation

The DSD is written $N(D)$ [mm$^{-1}$ m$^{-3}$], and is defined as the concentration in air of raindrops with equivolume diameter in the interval $[D, D + \delta D)$ mm. The equivolume diameter is used because raindrops become oblate with size (e.g. Thurai and



Bringi, 2005); it is simply the diameter of a sphere that contains the same volume of water as a drop. $M_n$ [mm$^n$ m$^{-3}$], the $n$th-order moment of the DSD, is

$$M_n = \int\limits_0^\infty N(D)D^n dD. \tag{1}$$

The double-moment normalisation method of Lee et al. (2004), allows for the DSD to be expressed as a combination of two
of its moments $M_i$ and $M_j$ of arbitrary orders $i$ and $j$, and a double-normalised DSD $h(x)$ [–], where $x = D M_i^{1/(j-i)} M_j^{-1/(j-i)}$
[-] is the second-normalised diameter (Lee et al., 2004). Using the normalisation, the DSD can be written

$$N(D) = M_i^{(j+1)/(j-i)} M_j^{(i+1)/(i-j)} h(x). \tag{2}$$

The method is flexible because the function $h(x)$ is not prescribed. Lee et al. (2004) showed that a generalised gamma model
is an appropriate choice for $h(x)$. Following their recommendation, we use the following double-normalised DSD (Lee et al.,
10 2004):

$$
\begin{aligned}
\hat{N}(D) &= M_i^{(j+1)/(j-i)} M_j^{(i+1)/(i-j)} \hat{h}(x), \\
\hat{h}(x) &= c\,\Gamma_i^{(j+c\mu)/(i-j)} \Gamma_j^{(-i-c\mu)/(i-j)} x^{c\mu-1} \\
&\quad \times \quad \exp\left[ -\left(\frac{\Gamma_i}{\Gamma_j}\right)^{c/(i-j)} x^c \right],
\end{aligned}
$$

(3)

(4)

where $\Gamma$ is the gamma function, $\Gamma_i = \Gamma(\mu+i/c)$ and $\Gamma_j = \Gamma(\mu+j/c)$, and $c$ [-] and $\mu$ [-] are parameters which must be fitted
to the normalised DSD model. Since this formulation allows any DSD to be described using only two of its statistical moments,
the task of our DSD-retrieval algorithm is to estimate two DSD moments from polarimetric radar data.

The question of whether the double-normalised DSD is invariant has been investigated. Compared to previous single-moment
normalisation approaches that vary by rainfall type (Sempere-Torres et al., 2000), the double-moment approach shows more
similarity across changes in rainfall type (Lee et al., 2004). Raupach and Berne (2016a) tested the normalised DSD across
spatial displacement and between different climatic regions, and showed that for practical purposes, the double-moment DSD
can be considered invariant across space. Lee et al. (2007) showed that $h(x)$ derived from time series measurements at one
location had low scatter around the average normalised DSD. In this study, we make the assumption that the double-moment
normalised DSD function $h(x)$ is invariant in space and time over the typical domain of interest. Thus, in the retrieval method
proposed here, the generalised gamma model parameters are static and variance in the DSD is explained through variance in
the two chosen DSD moments.





## 3 Bulk rainfall variables

All bulk rainfall variables can be derived from the DSD (a detailed review is provided by Bringi and Chandrasekar, 2001). The mass-weighted mean drop diameter $D_m$ [mm], useful as a characteristic drop size, is $M_4/M_3$. Liquid water content $W$ [g m$^{-3}$] is related to the third moment of the DSD and is written

$$W = \frac{\pi}{6} 10^{-3} \rho_w M_3, \tag{5}$$

where $\rho_w$ [g cm$^{-3}$] is the density of water. The rain rate $R$ [mm h$^{-1}$] is defined as

$$R = 6\pi 10^{-4} \int_0^\infty v(D) D^3 N(D) dD, \tag{6}$$

where $v(D)$ [m s$^{-1}$] is the still-air fall speed of a drop with equivolume diameter $D$. In this study $v(D)$ was calculated using the method of Beard (1976).

Radar properties can also be derived from the DSD. In Rayleigh scattering, when the radar wavelength is much larger than the particles being measured and drops are assumed to be spherical, the radar reflectivity is $Z = M_6$ (Marshall et al., 1947). In Mie scattering, in which the wavelength is of similar size to the particles, reflectivity at horizontal polarisation $Z_h$ [mm$^6$ m$^{-3}$] is defined as (Bringi and Chandrasekar, 2001)

$$Z_h = \frac{10^6 \lambda^4}{\pi^5 |K|^2} \int_0^\infty \sigma_{bh}(D) N(D) dD, \tag{7}$$

where $\lambda$ [cm] is the wavelength, $|K|^2$ [-] is the dielectric factor of water, and $\sigma_{bh}(D)$ [cm$^2$] is the back-scattering cross-section of a raindrop of equivolume diameter $D$. Reflectivity in vertical polarisation, $Z_v$ [mm$^6$ m$^{-3}$], is obtained by replacing $\sigma_{bh}(D)$ with the vertically polarised back-scattering cross-section $\sigma_{bv}(D)$ [cm$^2$]. It is usual practice to deal with radar reflectivities in dBZ. Non-linear reflectivities are calculated as $Z_H = 10 \log_{10} Z_h$ and $Z_V = 10 \log_{10} Z_v$. Differential reflectivity $Z_{DR}$ [dB] is $Z_H - Z_V$. Differential reflectivity in linear units, $\xi_{dr}$ [-], defined as $Z_h/Z_v$, has been shown to relate to the reflectivity-weighted mean drop axis ratio $r_z$ (Jameson, 1983). $r_z$ is defined as

$$r_z = \frac{\int_0^\infty r(D) D^6 N(D) dD}{\int_0^\infty D^6 N(D) dD}, \tag{8}$$





where $r(D)$ is the vertical to horizontal axis ratio of a drop of equivolume diameter $D$. The relationship found by Jameson (1983) is

$$r_z \sim (\xi_{\mathrm{dr}})^{-\frac{3}{7}}, \tag{9}$$

which is valid for narrow distributions of raindrop axis ratio (Bringi and Chandrasekar, 2001).

Dual-polarisation radars measure specific differential phase shift (on propagation) $K_{\mathrm{dp}}$ [° km$^{-1}$], which is the difference in phase change between horizontally and vertically polarised waves that pass through one kilometre of rain. It is defined as (Bringi and Chandrasekar, 2001)

$$K_{\mathrm{dp}} = \frac{180\lambda}{\pi} 10^{-1} \int\limits_0^\infty \mathrm{Re}\left[f_{hh}(D) - f_{vv}(D)\right] N(D) dD, \tag{10}$$

where Re represents the real part of a complex number and $\mathrm{Re}(f_{hh})$ [cm] and $\mathrm{Re}(f_{vv})$ [cm] are the real parts of the forward
scattering amplitudes for horizontal and vertical polarisation respectively. $K_{\mathrm{dp}}$ can also be defined from the DSD as (Jameson, 1985)

$$K_{\mathrm{dp}} = \left(\frac{180}{\lambda}\right) 10^{-1} CW(1 - r_m), \tag{11}$$

where the dimensionless value $C \sim 3.75$ (Bringi and Chandrasekar, 2001). $r_m$ is the mass-weighted mean raindrop axis ratio, defined as (Jameson, 1985)

$$r_m = \frac{\int\limits_0^\infty r(D)D^3 N(D) dD}{\int\limits_0^\infty D^3 N(D) dD}. \tag{12}$$

Various axis ratio functions are available (e.g. Pruppacher and Beard, 1970; Andsager et al., 1999; Brandes et al., 2002; Thurai and Bringi, 2005). Where unspecified, the ratio function used here was that of Thurai and Bringi (2005). We return to the question of axis ratios and $K_{\mathrm{dp}}$ in Section 5. The integrals in this section and Equation 1 are idealised because the range of drop sizes is written from zero to infinity. Using measured data, the integrals were calculated over truncated classes of diameter
and second-normalised diameter, with $D$ or $x$ as class centres and $dD$ and $dx$ as the class widths. Since truncation potentially effects bulk variables (e.g. Willis, 1984; Ulbrich, 1985; Vivekanandan et al., 2004), we used the same truncation limits for compared quantities. When polarimetric variables were calculated from DSDs, the T-matrix codes of Mishchenko and Travis (1998) were used to calculate raindrop scattering properties. Unless specified otherwise, the codes were used with an assumed temperature of 12.5° C, a Gaussian distribution of raindrop canting angles with zero mean and a standard deviation of 6°
(stated as reasonable by Bringi and Chandrasekar, 2001), and a radar frequency of 9.4 GHz.





## 4 Data

To test the new method, data from three networks of OTT Parsivel (Löffler-Mang and Joss, 2000) disdrometers were used. Each network had a nearby X-band weather radar that scanned above the disdrometers. These networks were also used in Raupach and Berne (2016a) and that study provides a full description of the data and their treatment, plus the coordinates for all stations.

Here we provide a summary of the data used. The first network provided the HyMeX data set. This network was located in Ardèche, France, in the autumns of 2012 and 2013, and was part of the Hydrological Cycle in the Mediterranean Experiment (HyMeX[1], Drobinski et al., 2014). In this study the test data included measurements from 11 disdrometers located in the 13 $\times$ 7 km$^2$ network. Also used were data from a METEK GmbH micro rain radar (MRR Peters et al., 2002, 2005; Tridon et al., 2011), within the network, which provided vertical profiles of DSD estimations recorded with 100 m vertical resolution and 10

s integration time. MXPol, a transportable Doppler dual-polarisation weather radar (for instrument details see Schneebeli et al., 2013) was located to the north-east of the disdrometer network. In 2013, MXPol recorded "stacked" plan position indicator (PPI) scans above the Parsivel network at elevations of four, five, six, eight, 10, 12, 14, 16, and 20 degrees above horizontal, with a return time of about six minutes. Six rainfall events in which the MRR and MXPol both recorded data were identified for 2013. The events were from 1.8 to 7.5 hours in length. Temperature data from a weather station at Pradel Grainage were

used to estimate the lowest freezing levels per event, assuming an atmospheric lapse rate of $-6.5°$ km$^{-1}$ (Wallace and Hobbs, 2006). The estimated freezing levels cutoff heights ranged from 1171 m to 2586 m above sea level, and only those MRR data from below the cutoff level per event were used. The best-performing Parsivel (by comparison with rain-gauges) was used at each location covered by a PPI scan. More network details and the list of identified events are provided in Raupach and Berne (2016a).

The HyMeX data set was used as the primary data set in this study: it was the set on which the technique was trained, and the only data set in which estimates of the DSD aloft were available. Two more data sets were used to test the technique in different climatologies, and on data on which the technique was not trained. The second instrument network was composed of five first-generation Parsivel disdrometers, and MXPol, in Payerne, Switzerland, and took measurements from February to July 2014. We used the MXPol PPI scan at five degrees above horizontal, which had a return time of about five minutes. The third data set

was from ten Parsivel[2] disdrometers (Petersen et al., 2014), in a network deployed in Iowa, United States, during the National Aeronautics and Space Administration (NASA) Iowa Flood Studies (IFloodS) Global Precipitation Mission (GPM) ground validation campaign. Overlooking this network was the University of Iowa's X-band radar XPOL5 (Mishra et al., 2016). We used PPI data recorded at three degrees above horizontal, with a return time of about eight minutes, for three days of heavy rainfall, the 25th, 26th, and 27th May 2013. The three networks were in regions with different climatologies (as described

in Wolfensberger et al., 2015). Table 1 provides a summary of the three campaigns and their instruments. Table 2 shows the instrument stations used here, the distance of each station to the PPI radar volumes used, and the number of radar scans that overlapped with one-minute observations.

---

[1]See http://www.hymex.org.



**Table 1.** Summary of the campaigns and instruments used in this study. Coordinates for Parsivel networks are bounding boxes of the region covered. Altitude is instrument altitude above sea level to nearest 10 m. Hours are provided only for non-instantaneous measurements, and show the total number of hours of data across all stations. For numbers of radar scans used, see Table 2.

| Data set | Instrument type | Coordinates | Altitude [m] | Hours |
|---|---|---|---|---|
| HyMeX | Parsivel (V1 and V2) | 44.5547 — 44.6141° N, 4.3826 — 4.5148° E | 200 — 640 | 398 |
| | MRR | 44.5790° N, 4.5011° E | 270 | 22 |
| | X-band radar | 44.6141° N, 4.5461° E | 600 | |
| Payerne | Parsivel (V1) | 46.8425 — 46.9783° N, 6.9184 — 7.13° E | 433 — 451 | 315 |
| | X-band radar | 46.8133° N, 6.9428° E | 489 | |
| Iowa | Parsivel (V2) | 41.64062 — 41.99267° N, 92.09138 — 91.54163° W | 197 — 286 | 333 |
| | X-band radar | 41.8870° N, 91.7341° W | 263 | |

Disdrometer data, which had raw integration times of either 30 s or 60 s, and MRR data were resampled to one-minute temporal resolution. HyMeX and Payerne Parsivel data were corrected with reference to a 2D-video-disdrometer (2DVD) (Raupach and Berne, 2015a, b). This procedure removed unrealistically large drops and those too far from expected velocities, and adjusted drop concentrations so that DSD moments more closely matched those of a collocated 2DVD. These Parsivel

data were also quality controlled so that only error-free time steps containing liquid precipitation were selected. Iowa Parsivel data were used as provided without further quality control. All available disdrometer and PPI data were used, while MRR data were subset to HyMeX event times so that likely solid precipitation was not considered. MRR data were attenuation-corrected (METEK, 2010; Peters et al., 2010) and contained DSDs retrieved with vertical wind ignored (Strauch, 1976; Peters et al., 2002), and negative concentrations (METEK, 2010) reset to zero. PPI radar reflectivities were compared to measurements

from disdrometers (and the MRR in HyMeX), and bias in $Z_H$ was corrected for each of the three locations. Bias in $Z_{DR}$ was estimated using vertical scans (birdbath scans, similar to Grazioli et al., 2015) in light rain, and was corrected in each of the three data sets. Two days of radar data from Payerne (2014-03-22 and 2014-04-08) exhibited higher radar bias and were not included in this study. Attenuation correction for the PPI data was performed using the ZPHI algorithm (Testud et al., 2000), and $K_{dp}$ was estimated using the method of Schneebeli et al. (2014). PPI scan data were sampled by taking the mean

values for radar volumes that overlapped horizontally the latitude and longitude of the point in question within the instrument's corresponding one-minute integration period. To discount noise, PPI values were subset to those for which $Z_H$ was greater than or equal to 10 dBZ and the signal to noise ratio in horizontal polarisation was greater than or equal to 5 dB. DSD data were treated as in Raupach and Berne (2016a): Parsivel DSDs were truncated to 0.2495 mm to 7 mm for all Parsivel data (Raupach and Berne, 2015a); to avoid including overestimated numbers of small drops (Peters et al., 2005), DSDs estimated by the MRR

were truncated to 0.6 mm to 5.8 mm and MRR data were further subset to records with $R \leq 150$ mm h$^{-1}$ (thus removing 0.5% of records); MRR data for altitudes greater than 2470 m were excluded because not enough points were available at those altitudes; and all DSDs were subset to time steps in which $R > 0.1$ mm h$^{-1}$.





**Table 2.** Instrument stations with corresponding PPI volumes, with the number of scans for that volume (S), and the volume centre's height above the ground (H (ground) [m], to nearest 10 m), the volume centre's height above sea level (H (a.s.l) [m], to nearest 10 m), and its horizontal range from the radar (D [km]). Also shown is the maximum one-minute rain intensity (MI [mm h$^{-1}$]) recorded by each Parsivel at a radar scan time.

| Network | Station | S | H (ground) | H (a.s.l) | D | MI |
|---|---|---|---|---|---|---|
| Payerne | HARAS Avenches | 481 | 910 | 1350 | 9.8 | 15.2 |
| | Military Airport Payerne | 406 | 370 | 820 | 3.7 | 16.3 |
| | Morat Airport | 326 | 2090 | 2520 | 23.2 | 16.6 |
| HyMeX | Lavilledieu | 1159 | 970 | 1190 | 8.4 | 53.4 |
| | Les Blaches | 1218 | 550 | 980 | 5.4 | 60.1 |
| | Lussas | 1234 | 730 | 1020 | 6.0 | 65.1 |
| | Mirabel | 1222 | 370 | 870 | 3.8 | 56.8 |
| | Mont-Redon | 1226 | 140 | 780 | 2.5 | 15.9 |
| | Pradel 1 | 1202 | 680 | 960 | 5.1 | 39.1 |
| | Pradel Grainage | 1179 | 700 | 970 | 5.3 | 42.8 |
| | Pradel-Vignes | 1185 | 730 | 990 | 5.5 | 22.0 |
| | Saint-Etienne-de-Fontbellon | 1008 | 1210 | 1520 | 13.1 | 40.7 |
| | St-Germain | 1089 | 1100 | 1310 | 10.1 | 73.4 |
| | Villeneuve-de-Berg | 1108 | 840 | 1140 | 7.7 | 62.0 |
| | Pradel Grainage (MRR) | 1179 | 700 — 1850 | 970 — 2120 | 5.3 | 97 |
| Iowa | apu05 | 85 | 1520 | 1810 | 29.5 | 44.5 |
| | apu06 | 84 | 1570 | 1840 | 30.1 | 52.0 |
| | apu07 | 80 | 1660 | 1930 | 31.9 | 38.1 |
| | apu08 | 87 | 1570 | 1850 | 30.3 | 59.7 |
| | apu09 | 158 | 700 | 940 | 12.9 | 40.0 |
| | apu10 | 164 | 640 | 890 | 12.0 | 22.9 |
| | apu11 | 155 | 600 | 860 | 11.4 | 20.8 |
| | apu12 | 154 | 540 | 801 | 10.3 | 27.8 |
| | apu13 | 96 | 1730 | 1920 | 31.7 | 60.1 |
| | apu14 | 95 | 1730 | 1920 | 31.7 | 65.2 |

## 5  DSD retrieval from polarimetric radar data

Raupach and Berne (2016a) showed that, for practical purposes, the double-normalised DSD can be assumed to be invariant across spacial displacement. Using this assumption, the DSD can be reconstructed at a point in space using polarimetric radar data. Given a known normalised DSD, the task of DSD reconstruction becomes that of determining from radar information the values of two DSD moments. In this section we present a new DSD-retrieval method that uses this idea. The aim of the proposed DSD-retrieval technique is to retrieve two DSD moments, using only polarimetric radar data. We used disdrometer data to simulate radar variables and estimate their relationships to DSD moments. The SCOP-ME method was trained with





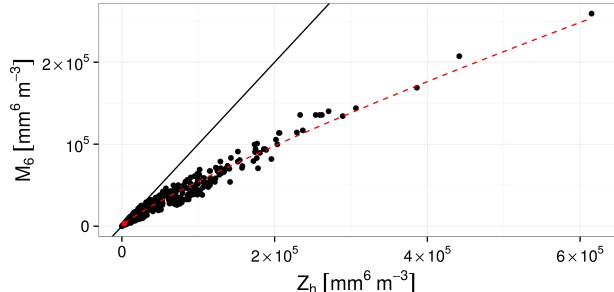

**Figure 1.** A sample of 20,000 points from the training set of Parsivel data, showing the relationship between radar reflectivity in linear units and moment six of the DSD. Each black point represents a one-minute DSD measurement. The one-to-one line is shown in black, and the red dashed line shows the fitted relationship of Equation 13. The low-high $Z_h$ threshold of 35 dBZ is shown with a red point.

DSDs simulated using a DSD model and a wide range of DSD parameter values. In contrast, we used empirical DSDs to train our method, to avoid any assumption about the shape of the DSD. A trade-off that must be made is that the measured DSDs are truncated. However, previous studies have shown that as long as the considered range of drop diameters is large enough around the median drop diameter, the affect of truncation on the calculated bulk variables is limited (Willis, 1984; Vivekanandan et al.,

2004). Willis (1984) concluded that the effect of maximum drop size truncation on bulk variables is negligible if the maximum considered drop size $D_{\max}$ is greater than $2.5D_0$. Using $D_0$ calculated from the recorded (truncated) DSDs, this criteria was met for 99.6% of the data we used for training. The criteria of Vivekanandan et al. (2004) is that, for there to be less than five percent error on bulk variables, the minimum drop size $D_{\min}$ should be less than $D_0/2$ and $D_{\max}$ should be greater than $4D_0$. This constraint was met by 92.5% of the DSDs we used (94.1% of the DSDs met this criteria for the upper drop size limit).

These two constraints were calculated assuming gamma DSDs. There remains the possibility that $D_0$ itself is subject to error because of the truncations, but we consider that these calculations give broad confidence in the bulk variables we used and the relationships between them. Further, the truncation used on our measured DSD data effects primarily very small drops, because drops larger than 7 mm are rare, and therefore its affect on the higher-order moments we use in the method is expected to be negligible. The disdrometer data used were the Parsivel data from the HyMeX network (101494 one-minute DSDs). $Z_H$,

$K_{\mathrm{dp}}$ and $Z_{\mathrm{DR}}$ were calculated for these DSDs for the MXPol stacked PPI incidence angles, temperatures of five, ten, and 15° C, and each of four drop axis ratio functions: those of Andsager et al. (1999), Brandes et al. (2002), Thurai and Bringi (2005), and that of Beard and Chuang (1987) in the form shown in Kalogiros et al. (2013).

    Radar reflectivity in linear units, $Z_h$ [mm$^6$ m$^{-3}$], is the sixth moment of the DSD in the Rayleigh scattering regime for spherical drops (Bringi and Chandrasekar, 2001). At X-band frequencies, larger drops enter into the Mie scattering regime and

differences appear between $M_6$ and $Z_h$. We use the observation that $Z_h$ departs from $M_6$ for heavier rain, and assume that this departure occurs when $Z_H$ is greater than some value. This threshold was determined through comparison of $M_6$ and $Z_h$ for



DSDs with $Z_H$ in classes of width 1 dBZ between 30 dBZ and 40 dBZ, and was set to 35 dBZ. For larger reflectivity values, a power law relationship was found using least-squares. The resulting relationship is

$$\widehat{M_6} = \begin{cases} Z_h & \text{if } 10\log_{10}(Z_h) \leq 35 \\ 3.2\,Z_h^{0.85} & \text{if } 10\log_{10}(Z_h) > 35. \end{cases} \tag{13}$$

On the training set, relative bias between $\widehat{M_6}$ and $M_6$ was $-2.7\%$, the interquartile range (IQR) of relative bias was 2.9 percentage points, and the $r^2$ value was 0.98. The fitted relationship is shown on samples of training data in Figure 1. Temperature made very little difference to the fitted parameter values (parameters fitted for the three individual temperatures tested were less than 1.5% different from those fitted to combined temperatures). Retrieving a second, lower-order DSD moment is more difficult than estimating $M_6$, because radar variables are more closely linked to the higher-order moments of the DSD. Using theoretical relationships as much as possible, we present here a method to retrieve the third moment of the DSD from polarimetric data.

As shown in Equation 9, the reflectivity-weighted mean drop axis ratio, $r_z$, is related to a negative power of the differential reflectivity in linear units, $\xi_{dr}$. In Kalogiros et al. (2013), the reflectivity-weighted and mass-weighted drop axis ratios were assumed to be the same and differences were dealt with through least-squares fitting of qualitative relationships between radar variables. A similar approach is taken here. Since $r_z$ and the mass-weighted mean drop axis ratio $r_m$ are both weighted mean drop axis ratios, we assume that $r_m$ can also be approximated through a negative power of $\xi_{dr}$, such that $r_m \sim \xi_{dr}^{-\beta_M}$. Recall from Equation 5 that $W$ is the third moment of the DSD multiplied by a constant. Replacing $W$, $r_m$, and constants in Equation 11 with $\hat{M}_3$ (to be estimated), $\xi_{dr}^{-\beta_M}$, and a constant $\alpha_M$ respectively, we arrive at the expression

$$\hat{M}_3 = \frac{\alpha_M K_{dp}}{(1 - \xi_{dr}^{-\beta_M})}, \tag{14}$$

where $\alpha_M$ and $\beta_M$ are parameters to be found. $K_{dp}$ is sensitive to the raindrop axis ratio (e.g. Bringi and Chandrasekar, 2001), so this relationship was found per axis ratio function using least-squares fitting. The same $Z_H$ threshold was used to divide the data into "Rayleigh-like" and "Mie-like" sets. The results and their performance statistics are shown in Table 3. The differences in fitted parameters for different assumed temperatures were generally low (fitted parameter values for the three individual temperatures tested were a maximum of 7% different from those fitted to combined temperatures), and the differences caused by different axis ratio function were much greater, depending on the functions compared. For this reason the parameters were fitted to the combined data for the three temperatures, per raindrop axis ratio function.

The proposed DSD retrieval technique can be summarised as follows: the double-normalised DSD $\hat{h}(x)$ and its parameters $c$ and $\mu$ are assumed trained from data and known. Then, given $K_{dp}$, $\xi_{dr}$ and $Z_h$, (1) DSD moment six is estimated using Equation 13; (2) DSD moment three is estimated using Equation 14 and parameter values from Table 3. The DSD is then retrieved using Equation 4 with $i = 3$ and $j = 6$.



**Table 3.** Fitted values of $\alpha_M$ and $\beta_M$ for DSDs with $Z_H \leq 35$ dBZ (Low $Z_H$) and those with $Z_H > 35$ dBZ (High $Z_H$), by drop axis ratio function (Ratio). Also shown are the resulting median relative bias (RB [%]), IQR of relative bias (IQR [% pts]), and $r^2$ on the training data.

| | Low $Z_H$ | | High $Z_H$ | | | | |
| Ratio | $\alpha_M$ | $\beta_M$ | $\alpha_M$ | $\beta_M$ | RB | IQR | $r^2$ |
| --- | --- | --- | --- | --- | --- | --- | --- |
| Thurai | 1524 | 6.8 | 730 | 2.5 | 2.5 | 13 | 0.97 |
| Brandes | 1332 | 4.3 | 684 | 2.0 | 1.4 | 23 | 0.96 |
| Andsager | 1623 | 6.1 | 803 | 2.5 | 0.6 | 20 | 0.96 |
| Beard | 1131 | 3.9 | 672 | 2.1 | 0.7 | 21 | 0.97 |

## 6 Comparison to an existing DSD-retrieval method

The new DSD retrieval method was compared to the DSD-retrieval method SCOP-ME (Anagnostou et al., 2009, 2010; Kalogiros et al., 2013). We implemented SCOP-ME using its description in Anagnostou et al. (2013). SCOP-ME was developed for X-band using simulated DSDs and T-matrix simulations of radar variables, and in Anagnostou et al. (2013) it is shown to

outperform the algorithms of Anagnostou et al. (2008) and Park et al. (2005a). The DSD model used by SCOP-ME is based on the normalised DSD of Willis (1984) (see also Bringi and Chandrasekar, 2001). Kalogiros et al. (2013) provided an explicit expression for rain rate using polarimetric variables, but since we are interested in the whole DSD, in the following sections we compare $R$ computed from reconstructed DSDs. The comparison of the two methods is first shown using Parsivel data in which the radar values were simulated using T-matrix codes and were therefore free of radar measurement noise.

Comparisons of the two techniques (both in this section and Section 8) were made using the three Parsivel data sets from HyMeX, Payerne, and Iowa. Comparison statistics were computed with difference defined as retrieved minus measured value, for DSD moments zero to seven, $D_m$, and $R$. For each one-minute DSD record, $Z_h$, $K_{dp}$ and $Z_{DR}$ were calculated using T-matrix codes, for an elevation angle of $4°$ above horizontal, and using each of the four drop axis ratio functions. SCOP-ME and the double-moment method were used to retrieve the DSD concentrations $N(D)$ for $D$ in the class centres of the truncated

Parsivel diameter classes. For the double-moment technique, the generalised gamma model parameters for the normalised DSD $\hat{h}(x)$ (Equation 4) for Parsivel data with $i = 3$ and $j = 6$ were used. As found by Raupach and Berne (2016a), these values were $c = 1.8$ and $\mu = 2.28$. The HyMeX data set is used as an example: measured and retrieved rain rates are shown for one event in Figure 2, for the drop axis ratio model of Beard and Chuang (1987). This model, which has been shown to match well to observations (Thurai et al., 2009), is shown because it provided the equilibrium drop shapes around which the SCOP-ME

training set was simulated (Kalogiros et al., 2013). Scatter plots of measured and reconstructed values for the Beard axis ratio model are shown in Figure 3. Comparisons of distributions of relative errors are shown in Figure 4.

Performance results are shown in detail for the HyMeX region in Table A1. Differences in performance statistic are shown by ratio function, variable, and region in Table A2. Differences were calculated as the absolute value of the double-moment technique metric minus the absolute values of the SCOP-ME metric. The metrics used were median relative bias, interquartile

range of relative bias, and distance from one of squared correlation, and distance from one of the slope of the best-fit line





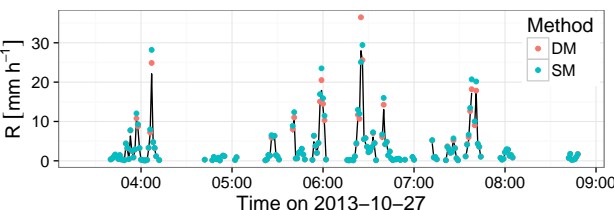

**Figure 2.** A time series plot showing retrieved versus measured rain rate for double-moment (DM) and SCOP-ME (SM) methods, for the fourth tested HyMeX event, using the axis ratio function of Beard and Chuang (1987). Measured rain rates by Parsivel are shown as a black line, retrieved values as coloured points.

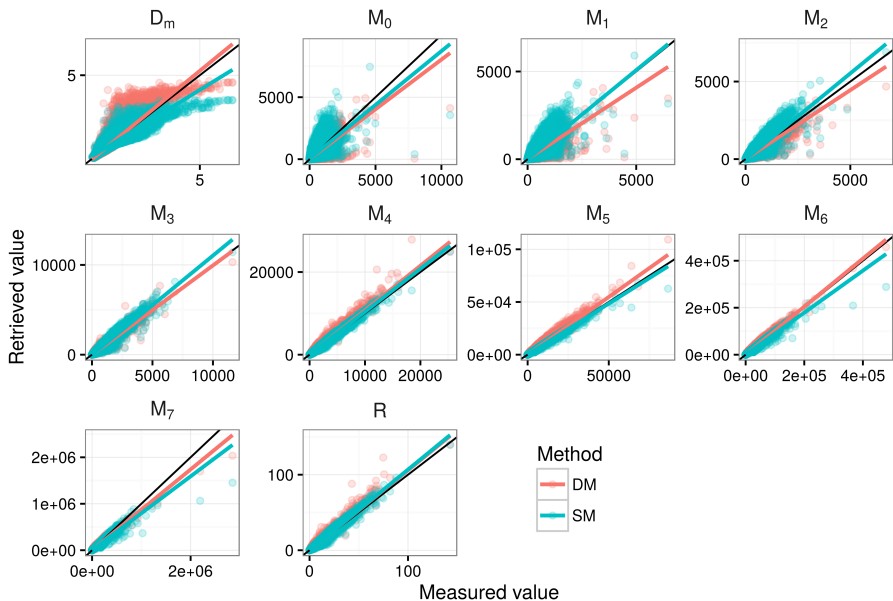

**Figure 3.** Scatter plots showing retrieved versus measured moments $M_n$ [$\mathrm{mm}^n\ \mathrm{m}^{-3}$], $R$ [$\mathrm{mm}\ \mathrm{h}^{-1}$], and $D_m$ [mm], for double-moment (DM) and SCOP-ME (SM) methods, on HyMeX data only, using the axis ratio function of Beard and Chuang (1987). One-to-one lines are shown in black, lines of best fit in colour.




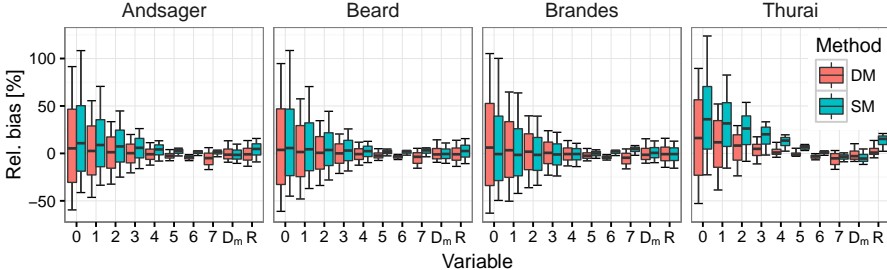

**Figure 4.** Relative bias distributions from the double-moment and SCOP-ME DSD-retrieval methods, by drop axis ratio function, for HyMeX data. Variables are moment order $n$ [mm$^n$ m$^{-3}$], $D_m$ [mm], and $R$ [mm h$^{-1}$]. Bold bars show medians, boxes show IQRs, whiskers show 10th to 90th percentile ranges.

on the scatter-plot of measured vs. reconstructed values. A negative value thus indicates that the double-moment technique performed better than SCOP-ME. These differences are shown visually in Figure 5. In over half of the tested region and variable combinations (moments one to seven, $R$, and $D_m$), the double-moment technique produced a better median relative bias than the SCOP-ME technique; on average the double-moment technique produced a median relative bias that was two percentage points better. IQR of relative bias was usually slightly higher for the double-moment technique, with a mean difference of 2.4 percentage points. Correlation coefficients and scatter plot slopes were usually very similar between the two techniques. On average, the double-moment results produced an $r^2$ value and slope that were respectively 0.02 and 0.03 further from one than the SCOP-ME values.

**Table 4.** Average differences between double-moment and SCOP-ME techniques, on Parsivel data, over three regions and four raindrop axis ratios.

| Variable | RB | IQR | $r^2$ | Slope |
|---|---|---|---|---|
| $D_m$ | -0.75 | 1.34 | 0.02 | -0.01 |
| $M_0$ | -2.31 | 12.32 | 0.05 | 0.06 |
| $M_1$ | -3.71 | 4.78 | 0.06 | 0.16 |
| $M_2$ | -4.22 | -0.08 | 0.01 | 0.04 |
| $M_3$ | -4.05 | -1.76 | 0.01 | -0.03 |
| $M_4$ | -3.03 | -0.14 | 0.02 | 0.09 |
| $M_5$ | -0.02 | 0.96 | 0.01 | 0.04 |
| $M_6$ | 1.22 | 0.09 | -0.00 | -0.11 |
| $M_7$ | 0.27 | 7.07 | -0.02 | -0.03 |
| $R$ | -3.35 | -0.25 | 0.02 | 0.07 |

The average differences across the three tested regions and four tested raindrop axis ratio functions are shown in Table 4. On average, the double-moment technique produced better median relative bias than SCOP-ME on $D_m$, $R$, and moments zero




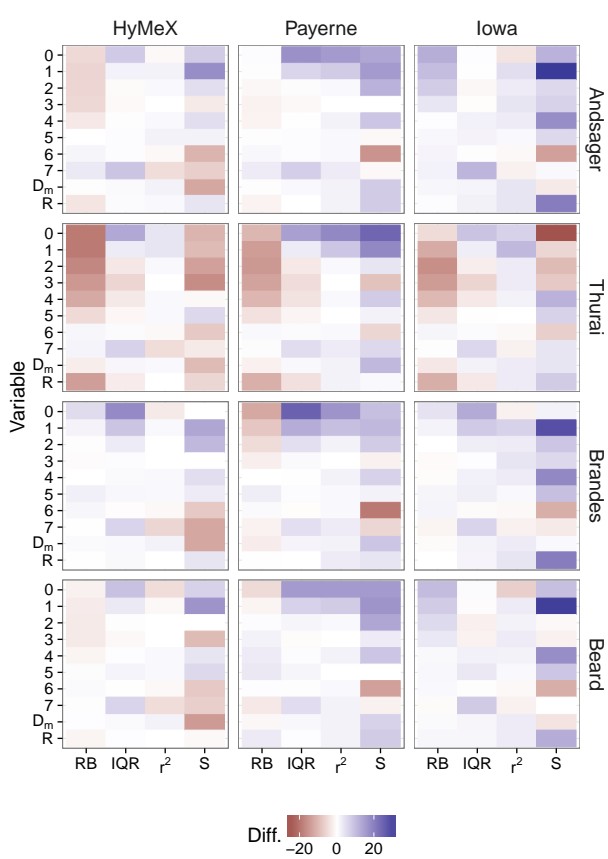

**Figure 5.** Differences in performance between the double-moment technique and SCOP-ME, using radar variables simulated from Parsivel data, by region and drop axis ratio function (differences in Tables A1 and A2). Reds indicate negative differences, where the double-moment technique outperformed SCOP-ME. Variables are moment order $n$ [mm$^n$ m$^{-3}$], $D_m$ [mm], and $R$ [mm h$^{-1}$]. Differences are shown for median relative bias (RB [% pts]), IQR of relative bias (IQR [% pts]), squared correlation coefficient ($r^2$, difference from one multiplied by 100 for display on the same scale), and scatter plot slope (S, difference from one multiplied by 100).





to five. IQRs were similar on average with the exception of moments zero and seven for which SCOP-ME produced smaller IQRs. The double-moment technique produced better scatter plot slope on $D_m$ and moments three, six and seven. As is shown in tables A1 and A2, the results were different for different drop axis ratio functions. For example, when the Thurai function was used, the double-moment technique performed better overall. In contrast, the performances of the two methods were more

similar when the Brandes function was used, and SCOP-ME outperformed the double-moment technique for moments zero and one in the Iowa data set with the Andsager and Beard axis ratio functions. The double-moment technique's performance changes by axis ratio function relate to the accuracy of the prediction of DSD moment three from $K_{dp}$ and $Z_{DR}$. As shown in Table 3, moment three is recovered most precisely when the Thurai axis ratio function is used. These results do not imply that the Thurai axis ratio function is more appropriate than others, but that the fitted relationships used in the proposed method

work best when these drop shapes are assumed. While differences exist between the results for the different regions, the inter-region differences in comparative performance of the two techniques were generally small. We now move to testing the two techniques on measured radar data, in which noise is a problem that must be dealt with.

## 7   Reducing the effects of noise

Radar data is noisy at light rain rates, particularly for $K_{dp}$ and $Z_{DR}$ (e.g. Bringi et al., 2002; Schneebeli et al., 2014). We

propose here a method to deal with this noise for the current application of DSD retrieval. Regressions on $Z_H$ and $\xi_{dr}$ are used to determine "expected" values for these variables, which can be used when the measured values are likely to be noisy. We found that $Z_{DR}$ can be reasonably predicted from $Z_h$ using

$$\hat{Z}_{DR} \sim \alpha_Z Z_h^{\beta_Z}, \tag{15}$$

and $K_{dp}$ can be predicted from $Z_h$ and $\xi_{dr}$ using

$$\hat{K}_{dp} \sim \alpha_K Z_h^{\beta_{K1}} \xi_{dr}^{\beta_{K2}}. \tag{16}$$

with parameters $\alpha_Z$, $\beta_Z$, $\alpha_K$, $\beta_{K1}$ and $\beta_{K2}$. Least-squares fitting in log-log space, using the training data set described in Section 5, was used to find best-fitting parameter values per raindrop axis ratio function. Just as for the retrieval of DSD moment six, assumed air temperature made only a small difference (parameter values fitted to individual temperature data sets differed by less than 3% from parameters values fitted using combined temperatures), whereas different axis ratios produced

more diverse parameter values. The resulting values and performance statistics are shown in Table 5.

We use threshold values based on those of Bringi et al. (2002) to determine when $K_{dp}$ and $Z_{DR}$ may be noisy. To reduce the effects of noise, then, if $Z_H \leq 35$ dBZ or $Z_{DR} \leq 0.2$ dB, measured $Z_{DR}$ is replaced by the the expected value $\hat{Z}_{DR}$ and $\xi_{dr}$ is replaced by $10^{(\hat{Z}_{DR}/10)}$. Likewise, if $Z_H \leq 35$ dBZ or $K_{dp} \leq 0.3$ ° km$^{-1}$, $K_{dp}$ is replaced by $\hat{K}_{dp}$ (calculated with $\hat{\xi}_{dr}$ if $\xi_{dr}$ was replaced). This method for treating radar data allows radar data with negative values of $K_{dp}$ or $Z_{DR}$ to be used. The data





**Table 5.** Fitted coefficients and the performance of the fits on disdrometer data with simulated polarimetric variables, for Equations 15 and 16. Performance is shown in terms of median relative bias (RB) and the IQR of relative bias (IQR).

| Ratio | $\alpha_Z$ | $\beta_Z$ | $Z_{\mathrm{DR}}$ performance RB [%] | IQR [% pts] | $\alpha_K$ | $\beta_{K1}$ | $\beta_{K2}$ | $K_{\mathrm{dp}}$ performance RB [%] | IQR [% pts] |
|---|---|---|---|---|---|---|---|---|---|
| Thurai | 0.10 | 0.27 | 0 | 45 | 0.0004 | 0.90 | -3.78 | 4 | 31 |
| Brandes | 0.03 | 0.41 | -3 | 74 | 0.0001 | 1.02 | -2.50 | -1 | 12 |
| Andsager | 0.05 | 0.35 | -2 | 59 | 0.0002 | 0.97 | -3.13 | -0 | 15 |
| Beard | 0.06 | 0.35 | -2 | 63 | 0.0002 | 1.00 | -3.14 | -2 | 12 |

treatment improved DSD-retrieval performance for both the double-moment and SCOP-ME techniques. For example, when retrieved DSDs were matched to measured MRR data, the median relative bias was reduced by an average (across variables) of ~10 percentage points for SCOP-ME and by ~18 percentage points for the double-moment technique, while IQRs were reduced more; for example on the MRR data the IQRs were reduced by ~92 (95) percentage points for the SCOP-ME (double-moment) method. When retrieved DSDs were compared to Parsivel data, the noise in the radar data contributed to errors to such an amount that for both techniques the proposed method for dealing with $K_{\mathrm{dp}}$ and $Z_{\mathrm{DR}}$ reduced both relative bias and IQR on relative bias on moments of orders three and lower by hundreds of percentage points. PPI data used in the following section were treated using this technique. We note that because most values of $Z_H$ recorded in the PPIs analysed here were lower than 35 dBZ, the majority of radar records were corrected this way.

## 8 Comparison using radar data

The DSD-retrieval techniques were applied to the three locations, using noise-corrected PPI radar data, so the retrieval techniques were evaluated independently of the noise-correction method. We used the elevation angles of the stacked PPIs for HyMeX, 5° for Payerne, and 3° for Iowa. Measured radar variables $Z_H$, $K_{\mathrm{dp}}$ and $Z_{\mathrm{DR}}$ were recovered for volumes corresponding to instrument locations. DSD retrieval was applied using these values, and the resulting DSDs compared to those that were measured by other instruments. Because the axis ratio of Thurai and Bringi (2005) produced the best results for the double-moment technique on the Parsivel data, the double-moment technique was used with parameters for the Thurai axis ratio function. Note that the assumption of axis ratio function affects only parameters of the double-moment technique, because the radar data used in this section are measured, not simulated, and the SCOP-ME technique is used as provided in Anagnostou et al. (2013). In the HyMeX campaign, the lowest available PPI elevation angle (4°) was used to compare results to Parsivels, but there was also an MRR at Pradel Grainage which retrieved estimates of the DSD aloft. MRR-derived DSDs were compared at nine different altitudes using the MXPol stacked PPIs above the HyMeX instrument network. All comparisons using PPI data involved a difference in measurement volume – a change-of-support problem that we expect will lead to greater error spread (e.g. Raupach and Berne, 2016b). We first address the comparisons with MRR for HyMeX, then move to the comparisons with the Parsivel networks in all three regions.





MXPol volume centre altitudes were projected into MRR altitude classes for comparison. The double-moment DSD-retrieval algorithm was used with the generalised gamma model $\hat{h}$ parameters (Equation 4) for MRR data and $i = 3$ and $j = 6$ found by Raupach and Berne (2016a), of $c = 0.57$ and $\mu = 29.6$. The reconstructed DSDs were made using classes of drop diameter from 0.6 to 5.8 mm with a class width of 0.1 mm, so that the truncation matched that of the MRR data. PPI values from eight

100 m altitude classes between about 900 and 2100 m above sea level, from 496 PPI scans, were compared to MRR estimates of the DSD aloft. Results of comparisons between MRR- and PPI-derived DSDs are shown for three example altitudes in Figure 6. There was good agreement between the recorded radar reflectivity recorded by both instruments, with a median relative bias of $-2\%$, an IQR on relative bias of 15 percentage points, and a value of $r^2$ of 0.66. Both techniques overestimated DSD moment orders zero to four and underestimated orders six and seven. Rain rate was recovered with a median relative

bias of 6% (IQR 92 % pts) by the double-moment technique and 21% (IQR 106 % pts) by SCOP-ME. The double-moment technique showed lower median relative bias than SCOP-ME on all variables except moments five and six, and smaller IQRs on all variables except $D_m$. Correlation coefficients were low for both techniques (the maximum $r^2$ was 0.32, by SCOP-ME for $D_m$), but the double-moment technique had a slightly higher value of $r^2$ in the majority of cases. High best-fit slopes (values from 1.6 to 4.6) were observed for moments five, six, and seven, and show the effect of a few outlier points in these

cases, which appeared in results for both techniques. Performance differences between the two techniques are shown in Table A3. Overall, the double-moment technique for DSD-retrieval out-performed SCOP-ME for the retrieval of DSD parameters and rain rate measured aloft by the MRR.

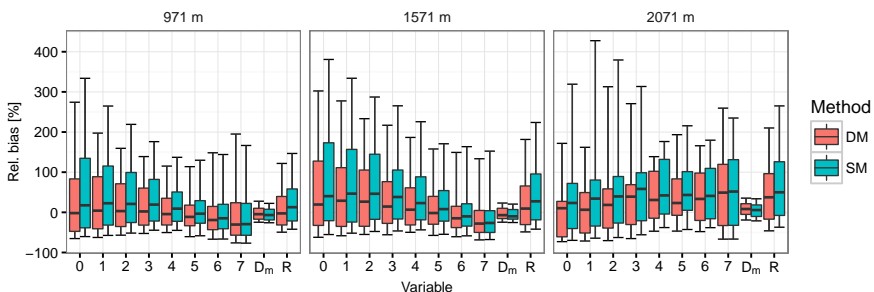

**Figure 6.** Distributions of relative bias on DSD moments orders between zero and seven, comparing DSDs retrieved using PPI data, and those measured by MRR at Pradel Grainage. The results are classed by altitude for a selection of three altitudes across the compared range. Symbols as for Figure 4.

DSDs retrieved from polarimetric radar data were also compared to those recorded by ground-based Parsivels in the three climatic regions we studied. It should be noted that there were, at times, significant vertical distances between the radar volume

and the instruments used in these comparisons (see Table 2). The DSDs were retrieved in truncated Parsivel drop diameter classes, using the Parsivel generalised gamma model parameters quoted in Section 6. Figure 7 shows distributions of DSD-retrieval relative error for each region.





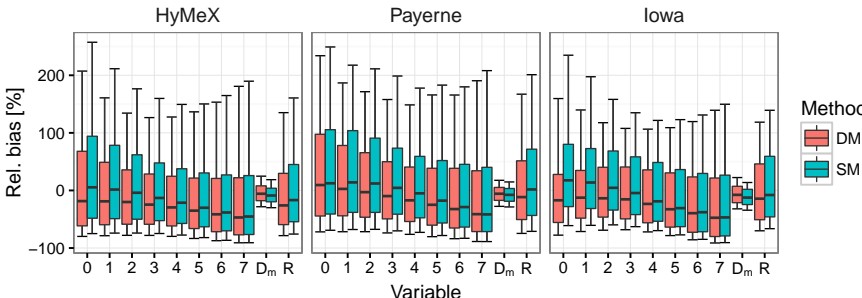

**Figure 7.** Distributions of relative bias on DSD moments orders between zero and seven, comparing DSDs retrieved using PPI data, and those measured by the Parsivel networks. Symbols as for Figure 4.

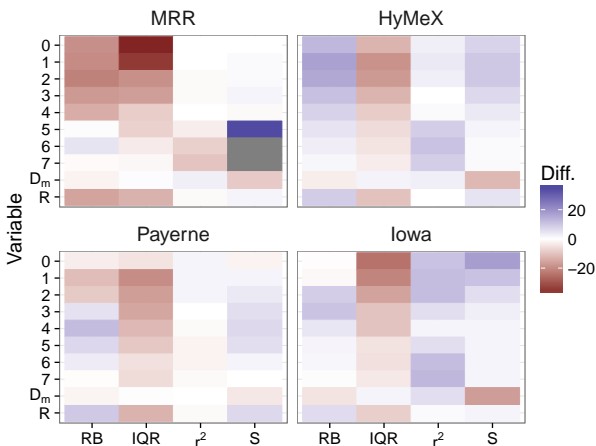

**Figure 8.** Differences in performance between the double-moment technique and SCOP-ME using noise-corrected radar data, for MRR and for Parsivels by region (differences in Table A3). Variables and performance statistics as for Figure 5. For display, difference in $r^2$ and slope are multiplied by 100. Red indicates that the double-moment technique outperformed SCOP-ME. Grey indicates an $r^2$ difference greater than 100 on this scale; these points were affected by scatter plot outliers.

In the majority of the tested cases, the double-moment technique produced smaller ranges of relative bias than SCOP-ME, for all variables except $D_m$. Where the double-moment technique produced better median relative bias, the mean per-case difference was $-4$ percentage points, while in cases where SCOP-ME performed better, the mean per-case difference was eight percentage points. Values of $r^2$ and scatter plot slope were similar between the two techniques, with the majority of cases showing differences less than 0.05 for both variables. Differences in performance between the two techniques are shown in Figure 8 and Table A3.





The performance of the double-moment DSD-retrieval technique is reliant on how accurately two DSD moments can be extracted from radar data, and in turn on how accurate the radar data are. Both retrieval techniques appear to be similarly affected by radar inaccuracies such as bias in $Z_H$, and experiments with different reflectivity bias corrections showed similar patterns of results. It is worth noting again that these comparisons were performed using data for which the noisy values of

$K_{dp}$ and $Z_{DR}$ had been treated using the method proposed in Section 7, which significantly improved the performance of both techniques with real radar data. The proposed DSD-retrieval technique was applied using the normalised DSD fitted to data in Ardèche, France, to the regions of Payerne in Switzerland and Iowa in the USA, without significant performance loss. This supports previous findings (Raupach and Berne, 2016a) that for practical use, the double-normalised DSD can be considered invariant.

## 9  Conclusions

Given the assumption of an invariant normalised DSD, and an estimate of that function, the DSD can be predicted using only two of its moments. Two DSD moments are available from polarimetric radar data. At X-band, radar reflectivity can be used to accurately predict the sixth moment of the DSD, and moment three can be retrieved relatively accurately using $K_{dp}$ and $\xi_{dr}$. We showed that by estimating these two moments from radar data, the DSD for a radar volume can be predicted using

the double-moment technique. Tests on disdrometer data from three networks in different climatic regions showed that DSD-retrieval using this new technique produced similar or slightly better performance than the SCOP-ME DSD-retrieval technique of Kalogiros et al. (2013). The proposed method is also more flexible, because there is no prescribed functional form for the double-normalised DSD, and even a non-parametric $\hat{h}(x)$ could be used. Nor is there a prescribed method of DSD moment extraction, which means that the moments used could be tailored to the intended purpose.

A new method for treatment of radar data with possibly noisy values of $K_{dp}$ and $Z_{DR}$ was proposed. The method is based on predicting the expected values of these variables from radar reflectivity, and considerably improved the performance of both the DSD-retrieval techniques. Using noise-corrected radar data, DSDs were predicted from polarimetric variables in PPI scans measured by X-band radars in each of the three regions. A test of the retrieved DSDs against MRR data for DSDs aloft in the HyMeX region in France, and comparisons of radar-retrieved DSDs against disdrometer data from the three regions,

showed reasonable agreement but large error spread for both methods. The double-moment technique predicted DSD moments measured by ground-based disdrometers with lower error spread than SCOP-ME. Compared to DSDs measured aloft by the MRR, the DSDs retrieved by the double-moment technique outperformed those of SCOP-ME. This study provides a proof-of-concept for DSD-retrieval using noise-corrected radar data, the double-moment normalisation method of (Lee et al., 2004), and a generalised gamma model for the normalised DSD. Future work will address more precise prediction of DSD moments

from polarimetric radar data.

*Acknowledgements.* For deploying and maintaining the instruments used, we thank, for HyMeX: J. Jaffrain, M. Schleiss, J. Grazioli, D. Wolfensberger, A. Studzinski (EPFL LTE), B. Boudevillain, G. Molinié, S. Gérard (Laboratoire d'étude des Transferts en Hydrologie et



Environnement (LTHE), Grenoble University), Y. Pointin and J. Van Baelen (Laboratoire de Météorologie Physique (LaMP), Université Blaise Pascal de Clermont-Ferrand). For Payerne: J. Jaffrain and Meteosuisse. We thank J. Grazioli for processing the MRR and MXPol data. HyMeX data were obtained from the HyMeX program, sponsored by grants MISTRALS/HyMeX, ANR-2011-BS56-027 FLOODSCALE project, OHMCV & EPFL-LTE. We thank the Swiss National Science Foundation for financial support under grant 2000021_140669.



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





**Table A1.** Comparison of double-moment method to SCOP-ME results on Parsivel data from five rainfall events in the HyMeX data set by axis ratio function (Ratio). Med. RB is median relative bias [%], RB IQR is interquartile range of relative bias [% points], $r^2$ is squared correlation coefficient. Slope is slope of best fit line on measured vs. reconstructed plot. Difference is difference in absolute values for RB and IQR, and difference in distance from 1 for $r^2$ and slope. A negative difference shows that the double-moment method improved on SCOP-ME's performance.

| Ratio | Var | Double-moment | | | | SCOP-ME | | | | Difference | | | |
|---|---|---|---|---|---|---|---|---|---|---|---|---|---|
| | | RB | IQR | $r^2$ | S | RB | IQR | $r^2$ | S | RB | IQR | $r^2$ | S |
| Ands. | $D_m$ | −1 | 10 | 0.85 | 1.05 | −2 | 9 | 0.87 | 0.82 | −0 | 1 | 0.02 | −0.13 |
| | $M_0$ | 5 | 77 | 0.67 | 0.80 | 11 | 69 | 0.65 | 0.89 | −5 | 8 | −0.01 | 0.08 |
| | $M_1$ | 3 | 52 | 0.77 | 0.81 | 9 | 50 | 0.79 | 1.01 | −6 | 2 | 0.02 | 0.18 |
| | $M_2$ | 1 | 33 | 0.89 | 0.88 | 7 | 34 | 0.90 | 1.07 | −6 | −1 | 0.01 | 0.05 |
| | $M_3$ | 0 | 20 | 0.96 | 0.98 | 6 | 21 | 0.96 | 1.05 | −6 | −1 | 0.00 | −0.03 |
| | $M_4$ | −1 | 11 | 0.98 | 1.07 | 4 | 11 | 0.99 | 0.98 | −3 | 0 | 0.01 | 0.05 |
| | $M_5$ | −3 | 5 | 0.98 | 1.09 | 3 | 4 | 0.99 | 0.93 | 0 | 0 | 0.02 | 0.02 |
| | $M_6$ | −4 | 3 | 0.98 | 1.02 | 2 | 3 | 0.97 | 0.87 | 2 | 0 | −0.01 | −0.11 |
| | $M_7$ | −5 | 12 | 0.98 | 0.86 | 2 | 3 | 0.93 | 0.79 | 3 | 9 | −0.05 | −0.07 |
| | $R$ | −1 | 12 | 0.98 | 1.05 | 5 | 12 | 0.98 | 1.01 | −4 | 1 | 0.01 | 0.04 |
| Thur. | $D_m$ | −3 | 9 | 0.88 | 1.07 | −5 | 7 | 0.89 | 0.84 | −3 | 1 | 0.01 | −0.10 |
| | $M_0$ | 16 | 80 | 0.70 | 0.97 | 36 | 66 | 0.74 | 1.15 | −20 | 14 | 0.04 | −0.11 |
| | $M_1$ | 12 | 49 | 0.80 | 0.89 | 32 | 46 | 0.84 | 1.21 | −20 | 3 | 0.04 | −0.10 |
| | $M_2$ | 8 | 27 | 0.91 | 0.91 | 26 | 30 | 0.92 | 1.22 | −18 | −4 | 0.01 | −0.14 |
| | $M_3$ | 5 | 12 | 0.98 | 1.00 | 20 | 18 | 0.97 | 1.17 | −15 | −6 | 0.00 | −0.17 |
| | $M_4$ | 1 | 5 | 0.99 | 1.07 | 14 | 9 | 0.99 | 1.08 | −13 | −3 | 0.01 | −0.01 |
| | $M_5$ | −1 | 3 | 0.98 | 1.07 | 7 | 4 | 0.99 | 0.99 | −5 | −1 | 0.01 | 0.06 |
| | $M_6$ | −3 | 3 | 0.99 | 0.98 | 2 | 2 | 0.97 | 0.90 | 1 | 1 | −0.01 | −0.08 |
| | $M_7$ | −5 | 12 | 0.98 | 0.82 | −4 | 5 | 0.93 | 0.79 | 2 | 7 | −0.05 | −0.03 |
| | $R$ | 1 | 6 | 0.99 | 1.05 | 15 | 9 | 0.99 | 1.12 | −14 | −3 | 0.00 | −0.06 |
| Bran. | $D_m$ | −1 | 12 | 0.83 | 1.06 | 1 | 11 | 0.86 | 0.81 | 1 | 1 | 0.02 | −0.13 |
| | $M_0$ | 6 | 87 | 0.63 | 0.79 | −1 | 68 | 0.59 | 0.80 | 5 | 19 | −0.03 | 0.00 |
| | $M_1$ | 3 | 59 | 0.74 | 0.79 | −2 | 50 | 0.75 | 0.93 | 2 | 9 | 0.01 | 0.14 |
| | $M_2$ | 2 | 38 | 0.87 | 0.86 | −2 | 35 | 0.87 | 1.02 | 0 | 3 | 0.00 | 0.11 |
| | $M_3$ | 1 | 23 | 0.96 | 0.97 | −1 | 22 | 0.95 | 1.03 | −1 | 0 | 0.00 | 0.00 |
| | $M_4$ | −1 | 13 | 0.98 | 1.07 | −1 | 12 | 0.99 | 0.98 | 0 | 1 | 0.01 | 0.05 |
| | $M_5$ | −2 | 6 | 0.98 | 1.11 | −0 | 5 | 0.99 | 0.92 | 2 | 1 | 0.01 | 0.03 |
| | $M_6$ | −4 | 3 | 0.98 | 1.05 | 2 | 3 | 0.97 | 0.87 | 1 | −0 | −0.01 | −0.08 |
| | $M_7$ | −5 | 11 | 0.98 | 0.91 | 5 | 5 | 0.92 | 0.78 | −0 | 7 | −0.06 | −0.13 |
| | $R$ | −1 | 14 | 0.98 | 1.04 | −1 | 13 | 0.98 | 1.00 | 0 | 1 | 0.00 | 0.04 |
| Beard | $D_m$ | −1 | 11 | 0.85 | 1.06 | −1 | 10 | 0.87 | 0.79 | 0 | 1 | 0.02 | −0.15 |
| | $M_0$ | 4 | 80 | 0.66 | 0.80 | 6 | 70 | 0.61 | 0.87 | −2 | 10 | −0.05 | 0.07 |
| | $M_1$ | 1 | 54 | 0.77 | 0.81 | 4 | 50 | 0.76 | 1.01 | −3 | 3 | −0.01 | 0.17 |
| | $M_2$ | 1 | 34 | 0.89 | 0.89 | 4 | 35 | 0.89 | 1.11 | −3 | −0 | 0.00 | 0.00 |
| | $M_3$ | 0 | 20 | 0.97 | 0.99 | 3 | 21 | 0.96 | 1.11 | −3 | −1 | 0.00 | −0.10 |
| | $M_4$ | −1 | 11 | 0.98 | 1.08 | 2 | 11 | 0.99 | 1.04 | −2 | 0 | 0.01 | 0.04 |
| | $M_5$ | −2 | 5 | 0.98 | 1.10 | 2 | 4 | 0.99 | 0.97 | 1 | 2 | 0.01 | 0.06 |
| | $M_6$ | −3 | 2 | 0.98 | 1.02 | 2 | 3 | 0.97 | 0.89 | 1 | −0 | −0.01 | −0.08 |
| | $M_7$ | −4 | 11 | 0.98 | 0.87 | 3 | 5 | 0.93 | 0.79 | 0 | 7 | −0.05 | −0.07 |
| | $R$ | −1 | 13 | 0.98 | 1.06 | 2 | 12 | 0.99 | 1.07 | −2 | 0 | 0.00 | −0.01 |





**Table A2.** Differences in performance by variable and region, for DSDs retrieved from Parsivel data using the double-moment technique and SCOP-ME. Differences are defined as for Table A1, so a negative difference shows that the double-moment method improved on SCOP-ME's performance. Note that differences for HyMeX are shown in Table A1.

| Ratio | order | Payerne | | | | Iowa | | | |
|---|---|---|---|---|---|---|---|---|---|
| | | RB | IQR | $r^2$ | S | RB | IQR | $r^2$ | S |
| Ands. | $D_m$ | 0 | 1 | 0.02 | 0.08 | 1 | 2 | 0.04 | −0.03 |
| | $M_0$ | 1 | 18 | 0.16 | 0.14 | 13 | 0 | −0.04 | 0.12 |
| | $M_1$ | −0 | 6 | 0.08 | 0.16 | 10 | −0 | 0.05 | 0.33 |
| | $M_2$ | −1 | 1 | 0.01 | 0.12 | 8 | −1 | 0.03 | 0.06 |
| | $M_3$ | −2 | −1 | 0.00 | 0.00 | 4 | −0 | 0.04 | 0.07 |
| | $M_4$ | −2 | 0 | 0.02 | 0.09 | 0 | 2 | 0.03 | 0.18 |
| | $M_5$ | −0 | 1 | 0.01 | −0.01 | 1 | 2 | 0.01 | 0.06 |
| | $M_6$ | 1 | 0 | 0.01 | −0.16 | 2 | 0 | −0.01 | −0.14 |
| | $M_7$ | 3 | 8 | 0.03 | −0.01 | 2 | 12 | −0.02 | 0.01 |
| | $R$ | −2 | 0 | 0.02 | 0.08 | 0 | 2 | 0.04 | 0.21 |
| Thur. | $D_m$ | −2 | 1 | 0.02 | 0.11 | −3 | 2 | 0.03 | 0.04 |
| | $M_0$ | −11 | 15 | 0.19 | 0.25 | −5 | 9 | 0.07 | −0.26 |
| | $M_1$ | −15 | 3 | 0.09 | 0.19 | −12 | 3 | 0.11 | −0.06 |
| | $M_2$ | −15 | −3 | 0.01 | 0.04 | −17 | −3 | 0.03 | −0.10 |
| | $M_3$ | −14 | −5 | 0.00 | −0.09 | −15 | −6 | 0.03 | −0.08 |
| | $M_4$ | −11 | −4 | 0.01 | 0.08 | −10 | −3 | 0.02 | 0.12 |
| | $M_5$ | −4 | −2 | 0.00 | 0.02 | −4 | 0 | 0.00 | 0.07 |
| | $M_6$ | 1 | 1 | 0.01 | −0.06 | 1 | 1 | −0.01 | −0.07 |
| | $M_7$ | 1 | 5 | 0.03 | 0.06 | −0 | 6 | −0.02 | 0.04 |
| | $R$ | −12 | −5 | 0.02 | 0.01 | −12 | −3 | 0.03 | 0.08 |
| Bran. | $D_m$ | −3 | 2 | 0.02 | 0.09 | −1 | 2 | 0.03 | 0.01 |
| | $M_0$ | −13 | 26 | 0.17 | 0.10 | 4 | 13 | −0.02 | 0.02 |
| | $M_1$ | −8 | 13 | 0.10 | 0.11 | 2 | 8 | 0.07 | 0.29 |
| | $M_2$ | −5 | 5 | 0.02 | 0.08 | 0 | 3 | 0.03 | 0.09 |
| | $M_3$ | −2 | 1 | 0.00 | −0.02 | −1 | 0 | 0.04 | 0.06 |
| | $M_4$ | −0 | 0 | 0.02 | 0.07 | −0 | 2 | 0.03 | 0.19 |
| | $M_5$ | 3 | 1 | 0.01 | 0.02 | 2 | 3 | 0.01 | 0.10 |
| | $M_6$ | 1 | −0 | 0.01 | −0.20 | 2 | −1 | −0.01 | −0.12 |
| | $M_7$ | −2 | 5 | 0.03 | −0.06 | −2 | 7 | −0.02 | −0.03 |
| | $R$ | −0 | −0 | 0.03 | 0.04 | −0 | 2 | 0.04 | 0.21 |
| Beard | $D_m$ | −1 | 1 | 0.02 | 0.07 | 1 | 1 | 0.03 | −0.04 |
| | $M_0$ | −5 | 16 | 0.16 | 0.16 | 10 | 1 | −0.07 | 0.10 |
| | $M_1$ | −2 | 7 | 0.08 | 0.17 | 8 | −1 | 0.03 | 0.32 |
| | $M_2$ | 1 | 2 | 0.01 | 0.14 | 6 | −2 | 0.02 | −0.01 |
| | $M_3$ | 2 | −0 | 0.00 | 0.03 | 4 | −2 | 0.03 | −0.02 |
| | $M_4$ | 3 | 1 | 0.02 | 0.09 | 1 | 2 | 0.02 | 0.18 |
| | $M_5$ | 4 | 1 | 0.00 | 0.00 | 1 | 3 | 0.01 | 0.09 |
| | $M_6$ | 0 | −0 | 0.01 | −0.14 | 1 | −0 | −0.01 | −0.12 |
| | $M_7$ | −3 | 5 | 0.02 | −0.02 | −1 | 8 | −0.02 | 0.00 |
| | $R$ | 3 | 0 | 0.02 | 0.08 | 1 | 1 | 0.03 | 0.13 |





**Table A3.** Differences in performance by variable and region, for DSDs retrieved from PPI data using the double-moment technique and SCOP-ME, compared to the MRR at Pradel Grainage (MRR) and Parsivels by region (HyMeX, Payerne, and Iowa). Differences are defined as for Table A2.

| Variable | | RB | IQR | $r^2$ | S |
|---|---|---|---|---|---|
| $D_m$ | MRR | −2 | 1 | 0.03 | −0.09 |
| | HyMeX | −3 | 2 | 0.03 | −0.12 |
| | Payerne | −2 | 0 | 0.00 | −0.04 |
| | Iowa | −5 | 2 | 0.06 | −0.17 |
| $M_0$ | MRR | −19 | −39 | 0.00 | 0.00 |
| | HyMeX | 13 | −13 | 0.03 | 0.08 |
| | Payerne | −3 | −5 | 0.02 | −0.02 |
| | Iowa | −0 | −25 | 0.11 | 0.18 |
| $M_1$ | MRR | −20 | −35 | 0.00 | 0.01 |
| | HyMeX | 17 | −19 | 0.04 | 0.10 |
| | Payerne | −11 | −20 | 0.02 | 0.02 |
| | Iowa | −1 | −21 | 0.12 | 0.11 |
| $M_2$ | MRR | −22 | −19 | −0.01 | 0.01 |
| | HyMeX | 16 | −18 | 0.03 | 0.10 |
| | Payerne | −9 | −17 | 0.02 | 0.04 |
| | Iowa | 9 | −16 | 0.12 | 0.06 |
| $M_3$ | MRR | −18 | −17 | −0.01 | 0.02 |
| | HyMeX | 12 | −13 | 0.00 | 0.07 |
| | Payerne | 5 | −15 | 0.00 | 0.06 |
| | Iowa | 11 | −10 | 0.06 | 0.03 |
| $M_4$ | MRR | −14 | −9 | 0.00 | −0.01 |
| | HyMeX | 8 | −9 | 0.01 | 0.04 |
| | Payerne | 12 | −12 | −0.01 | 0.07 |
| | Iowa | 4 | −10 | 0.02 | 0.02 |
| $M_5$ | MRR | 1 | −8 | −0.03 | 0.35 |
| | HyMeX | 5 | −6 | 0.09 | 0.02 |
| | Payerne | 7 | −10 | −0.02 | 0.06 |
| | Iowa | 2 | −5 | 0.06 | 0.02 |
| $M_6$ | MRR | 5 | −3 | −0.08 | 1.12 |
| | HyMeX | 3 | −5 | 0.11 | 0.01 |
| | Payerne | 3 | −5 | −0.02 | 0.02 |
| | Iowa | 2 | −5 | 0.12 | 0.02 |
| $M_7$ | MRR | −1 | −1 | −0.10 | 1.56 |
| | HyMeX | 2 | −3 | 0.09 | 0.01 |
| | Payerne | −0 | −6 | −0.01 | 0.00 |
| | Iowa | 1 | −4 | 0.13 | 0.02 |
| $R$ | MRR | −16 | −13 | −0.01 | 0.02 |
| | HyMeX | 9 | −10 | 0.00 | 0.05 |
| | Payerne | 10 | −13 | −0.01 | 0.07 |
| | Iowa | 6 | −8 | 0.01 | 0.02 |