# Peer review of "Retrieval of the raindrop size distribution from polarimetric radar data using double-moment normalisation"

_Atmospheric Measurement Techniques, 2016_

## Referee Comment (RC1) · Anonymous Referee #1 · 22 Dec 2016

Title: Retrieval of the raindrop size distribution from polarimetric radar data using double-moment normalization. Author(s): Timothy H. Raupach and Alexis Berne MS No.: amt-2016-301

In this paper, the authors present a new technique to estimate the raindrop size distribution and its parameters directly from polarimetric radar measurements. As already highlighted in the quick report, the present work can be of particular interest for the radar meteorologist scientific community. The logic flow of the conducted analysis is well exposed. The main revision points refer to the presentation of the results. The tables are very useful, while the figures are sometimes a bit small and it is difficult to easily distinguish dots, lines (i.e. in Figure 2 is difficult to distinguish the retrieved

and measured rain rate time series). I suggest to increase the figure size where it is possible. As general comment, at least in one out of the three datasets (it is not well explained if the HyMeX dataset is provided by Parsivel first or second generation), the authors use the Parsivel first generation data. Has been demonstrated the better performance of Parsivel second generation (Parsivel2) with respect to the first generation (Tokay et al., 2014 ,JTECH). Even if the authors, using the Raupach and Berne 2016a,b approach to correct the Parsivel data, this could affect the goodness of the results.

Considering these general and the following specific comments, I recommend the publication of the paper on the Atmospheric Measurement Techniques after the authors address the revision.

Specific comments.

- The simulation of the radar variable from disdrometric measurements to test the efficiency of the proposed technique with respect to the common used technique is particularly appreciated and useful. In the Section 5, they describe the difference between Rayleigh and Mie scattering region as function of the raindrop size at X-band frequency. They put a threshold at ZH=35 dBZ to discriminate the two regions by using the HyMeX Parsivel data only. What about the other two datasets? If they apply the say procedure, do they obtain the same threshold? There may be a climatic dependence on this threshold (i.e. the same reflectivity can be obtained by different DSD with a higher (lower) number of smaller (larger) drops respectively, which fall in the Rayleigh or Mie scattering region).

- From Figure 1 in linear scale, it is almost impossible to individuate ZH=35 dBZ (3.16e+03 mm6m-3). I suggest to change the linear to dBZ scale.

- Figures 2-4. I suggest to increase the size of the plots. Even the dots size (especially for Figure 3) can be slightly decreased to a better interpretation of the figures.

- Page 12. This is probably the most confusing part of the paper for the results interpretation. I clearly understand the summarize in only one figure the big amount of results is not easy, but some point arise reading this part. It could be useful for the reader, that the authors recall in the text the explanation of the Figure 5 and Table A1 and A2, which they give in the captions as well as the indicators used (relative bias, IQR of relative bias, correlation coefficient and slope of fit). They also show in Figure 5 the difference in performance between the double-moment technique and SCOPE-ME highlighting the cases where a method outperforms the other. On the other side, Table A1 reports also the absolute values of the considered indicators. I suggest to add this information at least in the Table A2. It is important to show which technique gives the best results, but it is equal (or more) important to know how far is the estimation from the measurement parameters.

- Page 8-line 13 and page 10-line 15: the authors say that they simulate the radar variables "for the MXPol stacked PPI incidence angles" and "for an elevation angle of 4°". Is the radar incidence angle really a input parameter in the T-matrix code? I retain that the incidence angle does not infer the simulation of the radar variables from disdrometer data. Please clarify this point.

- Page 13. Table 3 summarizes the performance difference combining all the Parsivel data and the four axis ratios used. The authors are combining data collected from "different" instruments (same physical base but different version). It could have more sense combining the data collected from the same instruments. Moreover, as they report in lines 5-8, the different axis ratio gives different results (with the Thurai function, the double-moment outperforms the SCOPE-ME, while the opposite is true when the Brandes function is used, etc.). Averaging over the axis ratios, there may be a sort of compensation in the results. My opinion is that could be more interesting just to show the difference for each axis ratio but averaging over the three regions (much better if the considered data are collected by the same instrument as already said). This could give an indication about a climatological dependence of the results.

- Figure 8 and Table A3: the results show that when the double-moment technique is applied to the radar data, the improvements with respect to the SCOPE-ME are not so evident as much as when the technique is applied to the radar variable as simulated from disdrometers. Can the authors tell something about this?

- Page 17-lines 2-4: please explain better the two sentences.

---

## Referee Comment (RC2) · Anonymous Referee #2 · 25 Jan 2017

General Comments

The authors generally replied satisfactorily to the comments on the original manuscript and added the relevant clarifications in the revision. Some points, which are described in specific comments below, need more explanation.

Specific Comments

Fig. 3: The authors didn't justify the large values (larger than 5mm) of measured Dm in Fig. 3, which are probably erroneous. Such large values come to clear contradiction with the note from authors in another comment on the effect of truncation limits of DSD on results that drops above 7 mm in diameter are rare. By excluding such unrealistic

[Figure]

large Dm values in Fig. 3 the correlation of the two estimation methods with measured values changes.

p. 15, l. 27-28: As it was mentioned in the comments on the original manuscript, the threshold of 35 dBZ for ZH to replace radar measured ZDR and KDP with expected values in order to avoid noise effects is too high. At X-band it corresponds on average to a value of 1.5 mm for Dm and values in ZDR and KDP higher than the corresponding thresholds of 0.2 dB and 0.3 deg/km, which they authors additionally use and are acceptable values. For example, the average relation at X-band between ZH and ZDR (Park et al. 2005, JTECH) shows that a value of 35 dBZ for ZH corresponds on average to 1.2 dB for ZDR, which is clearly a value that is above noise for all polarimetric radars.. A 15 dBZ threshold for ZH would be more realistic. The 35 dBZ threshold reported in the paper of Bringi et al. (2002) that the author use a reference for such a high value corresponds to S-band radar data (lower ZDR than X-band) and it used to discriminate light rain (usually stratiform) from more intense rain in order to use a different retrieval method in this case. Similar use for the 35 dBZ threshold is made by Part et al. (2005) in rainfall estimator (with or without KDP). This does not mean that 35 dBZ correspond to noisy ZDR or KDP in order to replace them with expected values, but simply that the specific polarimetric rainfall estimators fail below this threshold.

[Figure]

---

## Author Comment (AC1) · 20 Mar 2017

This document provides responses to reviews of our manuscript AMT-2016-301-RC1. A version of the manuscript with text changes highlighted is attached as a supplement to this comment (changes to references, tables, and removed text are not highlighted). The main changes that have been made are summarised as follows:

1. Some relevant references were added to the introduction.

2. Relationships between radar variables, and the parameters of the generalised gamma distributions used for the double-normalised DSD models, are now trained using all three data sets combined. The result is better performance of the

suggested technique on the Payerne and Iowa data sets, and a slight reduction in performance for low-order moments in the HyMeX data set. In accordance, instead of training on HyMeX data and applying the technique to the other data sets, we now split all Parsivel data into training (60%) and validation (40%) sets.

3. A second power-law fit for prediction of DSD moment six from radar reflectivity when $Z_H$ is low has been added, and both fits are now made using an orthogonal fitting procedure in log-log space.

4. The method for predicting moment three of the DSD has been updated for better accuracy. The updated method removes the requirement for a threshold on $Z_H$. In general, the new moment three and six predictions result in the DSD-retrieval technique performing better, especially for higher-order DSD moments.

5. The threshold value on $Z_H$ for the prediction of DSD moment six has been updated and better justified in response to reviewer comments.

6. Instead of using the raindrop axis ratio of Thurai and Bringi (2005), we use the newer relationship of Thurai et al. (2007) in its place. The performance of SCOP-ME is better with this newer axis ratio function, and therefore comparisons between the proposed and existing techniques are fairer.

7. We now include all available Parsivel instruments instead of using only the best-performing station when instruments are collocated.

In the following sections we address all reviewer comments and explain which changes were made in response to each one.

**1 Reviewer 1**

We thank reviewer 1 for the constructive comments, and respond to each one below.

1. **Reviewer:** *In this paper, the authors present a new technique to estimate the raindrop size distribution and its parameters directly from polarimetric radar measurements. As already highlighted in the quick report, the present work can be of particular interest for the radar meteorologist scientific community. The logic flow of the conducted analysis is well exposed. The main revision points refer to the presentation of the results. The tables are very useful, while the figures are sometimes a bit small and it is difficult to easily distinguish dots, lines (i.e. in Figure 2 is difficult to distinguish the retrieved and measured rain rate time series). I suggest to increase the figure size where it is possible.*

   **Response:** We have reviewed the size of the figures, their text and symbols.

   **Changes:** Where possible we have increased the figure size and adjusted symbol sizes. The changes made to each manuscript figure are:

   - Figure 1: Text size and symbol size have been increased, and the threshold point has been made clearer.
   - Figure 2 has been removed, since the SCOP-ME and double-moment technique results are close enough to be difficult to distinguish.
   - Figure 3 (now Figure 2): all combinations of data set and axis ratio function are now shown. Text size and figure size have been increased.
   - Figure 4 (now Figure 3): instead of a scatter plot we now show densities of measured vs. recovered values. Because it is not possible to overlay two densities, and the differences between the two techniques are best shown using the regression lines, we show the densities only for the double-

   moment technique. Regression lines are then shown for both techniques. The figure size and text size have been increased.
   - Figure 5 (now Figure 4): Text size has been increased.
   - Figure 6 (now Figure 5): Text size has been increased, and now we show results for SCOP-ME using raw and noise-corrected data.
   - Figure 7 (now Figure 6) and Figure 8 (now Figure 7): Text size has been increased.

2. **Reviewer:** *As general comment, at least in one out of the three datasets (it is not well explained if the HyMeX dataset is provided by Parsivel first or second generation), the authors use the Parsivel first generation data. Has been demonstrated the better performance of Parsivel second generation (Parsivel2) with respect to the first generation (Tokay et al., 2014 ,JTECH). Even if the authors, using the Raupach and Berne 2016a,b approach to correct the Parsivel data, this could affect the goodness of the results.*

   **Response:** We agree that information on the HyMeX network was missing, and that the Parsivel[2] provides better performance than the first-generation Parsivels. The HyMeX data set is a mixture of the first-generation and Parsivel[2], the Payerne data set is composed only of first-generation instruments, and the Iowa data set includes only Parsivel[2]. Our method is thus trained on a combination of first-generation and Parsivel[2] data. It is difficult to precisely determine the effect of our use of (corrected) first-generation Parsivel data on our results, but given that the technique is now trained on all three data sets combined we assume that the greater number of Parsivel[2] disdrometers now included will increase the representativeness of the trained approach.

   **Changes:** Specified the details about the HyMeX network and added a note about the limitations of Parsivel instruments to Section 4.

3. **Reviewer:** *The simulation of the radar variable from disdrometric measurements to test the efficiency of the proposed technique with respect to the common used technique is particularly appreciated and useful. In the Section 5, they describe the difference between Rayleigh and Mie scattering region as function of the raindrop size at X-band frequency. They put a threshold at ZH=35 dBZ to discriminate the two regions by using the HyMeX Parsivel data only. What about the other two datasets? If they apply the say procedure, do they obtain the same threshold? There may be a climatic dependence on this threshold (i.e. the same reflectivity can be obtained by different DSD with a higher (lower) number of smaller (larger) drops respectively, which fall in the Rayleigh or Mie scattering region).*

   **Response:** To further generalise our proposed method, we now train the technique using combined data from the three training sets (HyMeX, Payerne, and Iowa). Also, moment six of the DSD is now predicted from $Z_h$ on both sides of this threshold, instead of simply taking it to be equal to $Z_h$ under the threshold. To choose the threshold we compared relative bias, IQR of relative bias, and squared correlation $r^2$ between $Z_h$ [mm$^6$ m$^{-3}$] and $M_6$ by classes of $Z_H$ [dBZ] between 10 and 40 dBZ with a class width of 2 dBZ. $Z_h$ in all three regions departed from $M_6$ between 24-30 dBZ; HyMeX and Payerne data sets showed a drop in $r^2$ for the 24-26 dBZ class, while Iowa showed a sharp drop in $r^2$ in the 28-30 dBZ class. We used the threshold from the combined data, which showed a drop in $r^2$ at 28 dBZ. This particular threshold has been updated to 28 dBZ in the revised version of the paper.

   **Changes:** Threshold updated to 28 dBZ.

4. **Reviewer:** *From Figure 1 in linear scale, it is almost impossible to individuate*

   *ZH=35 dBZ (3.16e+03 mm$^6$ m$^{-3}$). I suggest to change the linear to dBZ scale.*

   **Response:** We agree that the threshold point was too difficult to distinguish.

   **Changes:** The plot has been changed to radar reflectivity in dBZ, and the threshold point has been updated to 28 dBZ indicated with a larger symbol in blue and white.

5. **Reviewer:** *Figures 2-4. I suggest to increase the size of the plots. Even the dots size (especially for Figure 3) can be slightly decreased to a better interpretation of the figures.*

   **Response:** We have reviewed the figures and increased their sizes, and reduced the size of the points in Figure 3 (now Figure 2). Please see the response to point 1 above for details about each figure.

6. **Reviewer:** *Page 12. This is probably the most confusing part of the paper for the results interpretation. I clearly understand the summarize in only one figure the big amount of results is not easy, but some point arise reading this part. It could be useful for the reader, that the authors recall in the text the explanation of the Figure 5 and Table A1 and A2, which they give in the captions as well as the indicators used (relative bias, IQR of relative bias, correlation coefficient and slope of fit). They also show in Figure 5 the difference in performance between the double-moment technique and SCOPE-ME highlighting the cases where a method outperforms the other. On the other side, Table A1 reports also the absolute values of the considered indicators. I suggest to add this information at least in the Table A2. It is important to show which technique gives the best results, but it is equal (or more) important to know how far is the estimation from the measurement parameters.*

**Response:** We agree on the importance of showing the performance of each technique as well as their differences. In the previous version of the paper we tried to limit the number of tables by showing only differences for Payerne and Iowa datasets. In response to this reviewer's comment we now include all results in the appendix, thus showing the performance of both techniques in all three regions. The tables are provided in an appendix since differences are summarised in the text and Figure 5 (now Figure 4). The metrics used are explained in the text in Section 6, as well as in the table and figure captions.

**Changes:** Updated explanations of Figure 5 (now Figure 4) and the performance statistics used. Replaced Table A2 with two tables of results, for the Payerne and Iowa data sets respectively, which become Tables A2 and A3. Table A4 now contains all performance statistics instead of only differences.

7. **Reviewer:** *Page 8-line 13 and page 10-line 15: the authors say that they simulate the radar variables "for the MXPol stacked PPI incidence angles" and "for an elevation angle of 4°". Is the radar incidence angle really a input parameter in the T-matrix code? I retain that the incidence angle does not infer the simulation of the radar variables from disdrometer data. Please clarify this point.*

**Response:** The incidence angle is an important input to the calculation of polarimetric radar variables from the DSD. As one example, $Z_{DR}$ at 90° (vertical incidence) is 0 dB because the reflectivity in horizontal and vertical polarisations is the same. At 0° incidence with larger raindrops present, the oblateness of the large drops is apparent and $Z_{DR}$ is larger than 0 dB.

8. **Reviewer:** *Page 13. Table 3 summarizes the performance difference combining all the Parsivel data and the four axis ratios used. The authors are combining*

*data collected from "different" instruments (same physical base but different version). It could have more sense combining the data collected from the same instruments. Moreover, as they report in lines 5-8, the different axis ratio gives different results (with the Thurai function, the double-moment outperforms the SCOPE-ME, while the opposite is true when the Brandes function is used, etc.). Averaging over the axis ratios, there may be a sort of compensation in the results. My opinion is that could be more interesting just to show the difference for each axis ratio but averaging over the three regions (much better if the considered data are collected by the same instrument as already said). This could give an indication about a climatological dependence of the results.*

**Response:** In Table 3 (now Table A4, the comparisons of DSDs retrieved from PPI data to MRR and Parsivel measured DSDs) there is no averaging over axis ratios. We chose to use one axis ratio that performed well (Thurai 2007) and use it for the double-moment technique in all PPI retrieval comparisons. Perhaps this was not made clear enough in the paper. We understand the reviewer's point about the different instrument types (or versions), but the two regions of Iowa and Payerne already split up the instruments into types, since the Iowa data was only Parsivel[2] and Payerne contained only first-generation Parsivel. The HyMeX data set contains both first-generation and Parsivel[2] disdrometers, but the correction procedure we apply to both is designed to make them more comparable (with reference to a 2DVD). The MRR data is always treated separately. Since we are not averaging over axis ratios, and since the two regions split the data into instrument types anyway, we prefer to leave the results in the same format.

**Changes:** Table A4 now contains not just differences but all performance statistics for both techniques, in order to respond to this reviewer's point 6.

9. **Reviewer:** *Figure 8 and Table A3: the results show that when the double-*

*moment technique is applied to the radar data, the improvements with respect to the SCOPE-ME are not so evident as much as when the technique is applied to the radar variable as simulated from disdrometers. Can the authors tell something about this?*

**Response:** There are a large number of other factors at play when the DSDs are retrieved from real PPI data, as compared to simulated radar variables from disdrometers. Using real radar data, there is the change of support problem that increases the error bar size, vertical distance between PPI-measured locations and ground-based instruments, and the noise in the radar data. All these factors combine to effect the performance of both DSD retrieval techniques, leading to greater uncertainty around the comparisons made using real radar data than those made using simulated radar variables from disdrometer data. This larger uncertainty tends to smooth out the differences between the two methods.

**Changes:** Notes about the vertical distance and change-of-support problem were both in the paper, but were in separate places; these have been put together in the introduction to Section 8, together with a note about the greater uncertainty when using PPI data.

10. **Reviewer:** *Page 17-lines 2-4: please explain better the two sentences.*

**Response:** These sentences related to the generalised gamma model parameters and drop size classes used for DSD retrieval to compare to MRR data.

**Changes:** The sentences have been re-written to include new details and are, we hope, clearer.

**2 Reviewer 2**

We thank reviewer 2 for the useful comments and we respond to each one below.

1. **Reviewer:** *Fig. 3: The authors didn't justify the large values (larger than 5mm) of measured Dm in Fig. 3, which are probably erroneous. Such large values come to clear contradiction with the note from authors in another comment on the effect of truncation limits of DSD on results that drops above 7 mm in diameter are rare. By excluding such unrealistic large Dm values in Fig. 3 the correlation of the two estimation methods with measured values changes.*

**Response:** We assume the reviewer was referring to Figure 4 (now Figure 3), the scatter plots. These large values of $D_m$ are very rare. While rare, such values do arise when using empirical and not modelled DSDs, even when the DSDs are truncated at 7 mm. In the Parsivel data sets, there were 0.018%, 0.005%, and 0.014% of $D_m$ values above 5 mm in the HyMex, Payerne, and Iowa data sets respectively. These DSDs passed our quality control procedures and therefore we have no reason to remove them; we thus leave them in the analyses.

**Changes:** A note about the rarity of these large values of $D_m$ has been added to the caption for Figure 4 (now Figure 3).

2. **Reviewer:** *p. 15, l. 27-28: As it was mentioned in the comments on the original manuscript, the threshold of 35 dBZ for ZH to replace radar measured ZDR and KDP with expected values in order to avoid noise effects is too high. At X-band it corresponds on average to a value of 1.5 mm for Dm and values in ZDR and KDP higher than the corresponding thresholds of 0.2 dB and 0.3 deg/km, which they authors additionally use and are acceptable values. For example, the average*

*relation at X-band between ZH and ZDR (Park et al. 2005, JTECH) shows that a value of 35 dBZ for ZH corresponds on average to 1.2 dB for ZDR, which is clearly a value that is above noise for all polarimetric radars.. A 15 dBZ threshold for ZH would be more realistic. The 35 dBZ threshold reported in the paper of Bringi et al. (2002) that the author use a reference for such a high value corresponds to S-band radar data (lower ZDR than X-band) and it used to discriminate light rain (usually stratiform) from more intense rain in order to use a different retrieval method in this case. Similar use for the 35 dBZ threshold is made by Part et al. (2005) in rainfall estimator (with or without KDP). This does not mean that 35 dBZ correspond to noisy ZDR or KDP in order to replace them with expected values, but simply that the specific polarimetric rainfall estimators fail below this threshold.*

**Response:** We thank the reviewer for this helpful comment. We have re-examined the choice of thresholds we use for $Z_H$ when deciding whether to replace possibly noisy values of $Z_{DR}$ and $K_{dp}$, with reference to this comment and the previous paper mentioned (Park et al., 2005). We have concluded that although the reviewer is correct that a value of $Z_H = 35$ corresponds to a higher value of $Z_{DR}$ than 0.2 dB, in the real radar data we used there are so many noisy values of $Z_{DR}$ and $K_{dp}$ below about 37 dBZ that the noise correction still needs to be applied in this range of $Z_H$ values.

Looking first at $Z_{DR}$, Fig. 1 of this comment shows the relationship between $Z_H$ and $Z_{DR}$ in the training data set we use, with a dotted line shows $Z_{DR} = 0.2$ dB. Horizontal lines show medians, vertical lines show 10th to 90th quantile ranges, boxes show interquartile ranges. There are differences between the relationship found here and that shown in Park et al. (2005) Figure 5, which we hypothesise are due to the differences in disdrometer used (including the correction we apply to some of our DSD measurements which tends to reduce the concentrations of small drops) and the climatology (Europe/USA vs. Japan). However, the plot

shows that if we assume (as per Bringi et al. (2002)) that $Z_{DR} = 0.2$ dB is a reasonable noise threshold, then 15-18 dBZ is a reasonable equivalent value for $Z_H$ at X-band, as the reviewer suggests. In our training data, the median value of $Z_H$ for values of $Z_{DR}$ between 0.19 and 0.21 is 18 dBZ. However, in real radar data $Z_{DR}$ is noisy for values of $Z_H$ below about 35-37 dBZ, as shown in Fig. 2 of this comment, which uses PPI data from the three studied regions and shows outlier points as dots. We therefore keep the $Z_H$ threshold at a higher value, and use 37 dBZ. The result of the correction on $Z_{DR}$ is shown in Fig 3 here, and the cleaned $Z_{DR}$ values are clearly closer to the theoretical values shown above.

Regarding $K_{dp}$, Fig. 4 of this comment shows the relationship between $K_{dp}$ and $Z_H$ in our training data. A dotted line showing $K_{dp} = 0.3$ ° km$^{-1}$. Again, horizontal lines show medians, vertical lines show 10th to 90th quantile ranges, boxes show interquartile ranges. The y-axis is in logarithmic scale to better distinguish $K_{dp} = 0.3$ ° km$^{-1}$. In this case, our threshold of $Z_H = 35$ dBZ seems reasonable, although some values of $K_{dp} < 0.3$ ° km$^{-1}$ fall into the class containing $Z_H$ up to 42.5 dBZ. For $K_{dp}$ between 0.29 and 0.31 ° km$^{-1}$ in our training data, the median value of $Z_H$ is 36.4 dBZ. To ensure we treat most noisy values we use an updated threshold value of 37 dBZ, which matches the threshold used for $Z_{DR}$. For $36.99 < Z_H < 37.01$ in our training data, the mean and median values of $K_{dp}$ are both 0.32 ° km$^{-1}$. Fig. 5 of this comment shows the observed PPI values for $Z_H$ vs. $K_{dp}$, in which noise is observed below about 37 dBZ. The noise-treated values of $K_{dp}$ are shown in Fig. 6 here. The treated values more closely match the theoretical relationship expected.

**Changes:** We use an updated $Z_H = 37$ dBZ threshold for treatment of noisy data. We are aware that changing the input data may change the relationships between observed radar variables, and therefore unduly penalise the SCOP-ME technique. To be fair in our comparisons we show the Parsivel results with no

noise cleaning, and we also show the difference made by the noise cleaning in the comparison with the MRR data in Figure 5 in the article. For the other results we show performance statistics for the techniques using cleaned radar data.

[Figure]

**Fig. 1.** Horizontal reflectivity to differential reflectivity relationship.

[Figure]

**Fig. 2.** Horizontal reflectivity to differential reflectivity measured relationship.

[Figure]

**Fig. 3.** Horizontal reflectivity to differential reflectivity relationship in cleaned radar data.

[Figure]

**Fig. 4.** Horizontal reflectivity to specific differential phase relationship.

[Figure]

**Fig. 5.** Horizontal reflectivity to specific differential phase measured relationship.

[Figure]

**Fig. 6.** Horizontal reflectivity to specific differential phase relationship in cleaned data.

**Supplement:**

[revised manuscript text omitted]

**3 Bulk rainfall variables**

All bulk rainfall variables can be derived from the DSD (a detailed review is provided by Bringi and Chandrasekar, 2001). The mass-weighted mean drop diameter $D_m$ [mm], useful as a characteristic drop size, is $M_4/M_3$. Liquid water content $W$ [g m$^{-3}$] is related to the third moment of the DSD and is written

$$W = \frac{\pi}{6} 10^{-3} \rho_w M_3, \tag{5}$$

where $\rho_w$ [g cm$^{-3}$] is the density of water. The rain rate $R$ [mm h$^{-1}$] is defined as

$$R = 6\pi 10^{-4} \int_0^\infty v(D) D^3 N(D) dD, \tag{6}$$

where $v(D)$ [m s$^{-1}$] is the still-air terminal fall speed of a drop with equivolume diameter $D$. In this study $v(D)$ was calculated using the method of Beard (1976), for site-specific altitudes and latitudes, and an assumed sea-level temperature of 15° and relative humidity of 0.95.

Radar variables can also be derived from the DSD. In Rayleigh scattering, when the radar wavelength is much larger than the particles being measured and drops are assumed to be spherical, the radar reflectivity is $Z = M_6$ (Marshall et al., 1947). In Mie scattering, in which the wavelength is of similar size to the particles, reflectivity in horizontal polarisation $Z_h$ [mm$^6$ m$^{-3}$] is defined as (Bringi and Chandrasekar, 2001)

$$Z_h = \frac{10^6 \lambda^4}{\pi^5 |K|^2} \int_0^\infty \sigma_{bh}(D) N(D) dD, \tag{7}$$

where $\lambda$ [cm] is the wavelength, $|K|^2$ [-] is the dielectric factor of water, and $\sigma_{bh}(D)$ [cm$^2$] is the back-scattering cross-section at horizontal polarisation of a raindrop of equivolume diameter $D$. Reflectivity in vertical polarisation, $Z_v$ [mm$^6$ m$^{-3}$], is obtained by replacing $\sigma_{bh}(D)$ with the vertically polarised back-scattering cross-section $\sigma_{bv}(D)$ [cm$^2$]. It is usual practice to deal with radar reflectivities in dBZ, calculated as $Z_H = 10 \log_{10} Z_h$ and $Z_V = 10 \log_{10} Z_v$.

Differential reflectivity $Z_{\mathrm{DR}}$ [dB] is $Z_H - Z_V$. Differential reflectivity in linear units, $\xi_{\mathrm{dr}}$ [-], defined as $Z_h/Z_v$, has been shown to relate to the reflectivity-weighted mean drop axis ratio $r_z$ [-] (Jameson, 1983). $r_z$ is defined as

$$r_z = \frac{\int_0^\infty r(D) D^6 N(D) dD}{\int_0^\infty D^6 N(D) dD}, \tag{8}$$

where $r(D)$ is the vertical to horizontal axis ratio of a drop of equivolume diameter $D$. The relationship found by Jameson (1983) is

$$r_z \sim (\xi_{\mathrm{dr}})^{-\frac{3}{7}}, \tag{9}$$

which is valid for narrow distributions of raindrop axis ratio (Bringi and Chandrasekar, 2001).

5    Dual-polarisation radars measure specific differential phase shift (on propagation) $K_{\mathrm{dp}}$ [° km$^{-1}$], which is the difference in phase produced between horizontally and vertically polarised waves that pass through rain. It is defined as (Bringi and Chandrasekar, 2001)

$$K_{\mathrm{dp}} = \frac{180\lambda}{\pi} 10^{-1} \int\limits_{0}^{\infty} \mathrm{Re}\left[f_{hh}(D) - f_{vv}(D)\right] N(D)dD, \tag{10}$$

where Re represents the real part of a complex number and $\mathrm{Re}(f_{hh})$ [cm] and $\mathrm{Re}(f_{vv})$ [cm] are the real parts of the forward
10  scattering amplitudes for horizontal and vertical polarisation respectively. Jameson (1985) showed that $K_{\mathrm{dp}}$ can be linked to the product of liquid water content and the deviation from unity of the mass-weighted mean raindrop axis ratio $r_m$ [-]. $r_m$ is defined as

[revised manuscript text omitted]

Disdrometer data, which had raw integration times of either 30 s or 60 s, and MRR data with 10 s integration time, were resampled to one-minute temporal resolution. HyMeX and Payerne Parsivel data were corrected with reference to 2D-video-disdrometer (2DVD) measurements from the HyMeX campaign (Raupach and Berne, 2015a, b). This procedure removed unrealistically large drops and those too far from expected velocities, adjusted velocity measurements, and adjusted drop concentrations so that DSD moments more closely matched those of the 2DVD. These Parsivel data were quality controlled so that only error-free time steps containing liquid precipitation were used. Iowa Parsivel data were used as provided without further quality control.

Parsivel data are subject to uncertainty due to differences across individual instruments and instrument generations (e.g. Jaffrain and Berne, 2011; Tokay et al., 2014; Thurai et al., 2011; Raupach and Berne, 2015a), and their limited sampling area introduces a bias, as reported by Tapiador et al. (2017). The Iowa data were provided in diameter class definitions that differed from those of the instrument manufacturer (Petersen et al., 2014). The HyMeX and Payerne data sets used the manufacturer's diameter class definitions, which implies the assumption of a raindrop axis ratio to equivolume diameter relationship (Battaglia et al., 2010). Our tests (not shown) showed limited differences made to DSD bulk variables when different axis ratio functions were used to modify the class definitions. Given the uncertainties involved in using modified diameter classes, we decided to use the manufacturer's class definitions for these two data sets. For each of the three regions, the Parsivel data were randomly sampled so that 60% of records formed a training data set and the remaining 40% formed an independent validation data set. Sensitivity of the random sampling was evaluated through repeated tests with different sample realisations and was found to be low.

All available disdrometer and PPI data were used, while MRR data were subset to event times so that likely solid precipitation was not considered. MRR data were attenuation-corrected (METEK, 2010; Peters et al., 2010) and contained DSDs retrieved with vertical wind ignored (Strauch, 1976; Peters et al., 2002). Negative concentrations (METEK, 2010) in MRR DSDs were

**Table 2.** Instrument stations with corresponding PPI volumes, with the number of scans for that volume (S), the volume centre's height above the ground (H (ground) [m], to nearest 10 m), height above sea level (H (a.s.l) [m], to nearest 10 m), and horizontal range from the radar (D [km]). MI [mm h$^{-1}$] is the maximum one-minute rain intensity recorded by each instrument at a radar scan time.

| Network | Station | S | H (ground) | H (a.s.l) | D | MI |
|---------|---------|---|-----------|-----------|---|-----|
| Payerne | HARAS Avenches | 483 | 914 | 1349 | 9.8 | 15.6 |
| | Military Airport Payerne | 408 | 365 | 816 | 3.7 | 16.7 |
| | Morat Airport | 349 | 2087 | 2520 | 23.2 | 16.9 |
| HyMeX | Lavilledieu | 1209 | 965 | 1192 | 8.4 | 55.5 |
| | Les Blaches | 1256 | 549 | 978 | 5.4 | 62.3 |
| | Lussas | 1277 | 732 | 1021 | 6.0 | 67.6 |
| | Mirabel | 1254 | 374 | 870 | 3.8 | 59.3 |
| | Mont-Redon | 1267 | 139 | 775 | 2.5 | 18.7 |
| | Pradel 1 | 1239 | 682 | 960 | 5.1 | 40.8 |
| | Pradel 2 | 1239 | 682 | 960 | 5.1 | 36.3 |
| | Pradel Grainage | 1216 | 700 | 971 | 5.3 | 44.6 |
| | Pradel-Grainage-v2 | 1216 | 700 | 971 | 5.3 | 45.1 |
| | Pradel-Vignes | 1222 | 733 | 989 | 5.5 | 22.7 |
| | Saint-Etienne-de-Fontbellon | 1099 | 1214 | 1516 | 13.1 | 53.6 |
| | St-Germain | 1139 | 1103 | 1307 | 10.1 | 76.2 |
| | Villeneuve-de-Berg | 1150 | 841 | 1142 | 7.7 | 84.0 |
| | Villeneuve-de-Berg 2 | 1152 | 841 | 1142 | 7.7 | 74.2 |
| | Villeneuve-de-Berg 3 | 1150 | 840 | 1141 | 7.7 | 72.5 |
| | Pradel Grainage (MRR) | 694 | 700 — 1850 | 970 — 2120 | 5.3 | 97 |
| Iowa | apu05 | 94 | 1522 | 1808 | 29.5 | 49.0 |
| | apu06 | 88 | 1566 | 1840 | 30.1 | 49.0 |
| | apu07 | 84 | 1661 | 1933 | 31.9 | 47.6 |
| | apu08 | 91 | 1569 | 1851 | 30.3 | 50.7 |
| | apu09 | 110 | 698 | 938 | 12.9 | 31.4 |
| | apu10 | 112 | 635 | 890 | 12.0 | 25.1 |
| | apu11 | 103 | 600 | 859 | 11.4 | 25.9 |
| | apu12 | 97 | 543 | 801 | 10.3 | 57.0 |
| | apu13 | 102 | 1727 | 1924 | 31.7 | 65.2 |
| | apu14 | 100 | 1727 | 1924 | 31.7 | 71.9 |

reset to zero. PPI radar reflectivities were compared to measurements from disdrometers (and the MRR in HyMeX), and bias in $Z_H$ was corrected on a per-campaign basis. Bias in $Z_{DR}$ was estimated using vertical scans (birdbath scans, similar to Grazioli et al., 2015), and was corrected in each of the three data sets. Two days of radar data from Payerne (2014-03-22 and 2014-04-08) exhibited higher radar bias due to hardware problems, and were not included in this study. Attenuation correction for the PPI data was performed using the ZPHI algorithm (Testud et al., 2000), and $K_{dp}$ was estimated using the method of

Schneebeli et al. (2014). PPI scan data were sampled for instrument locations by taking the mean values of radar volumes that overlapped horizontally the instrument coordinates within the instrument's corresponding one-minute integration period. To discount noise, PPI records were subset to those for which $Z_H$ was greater than or equal to 10 dBZ and the signal to noise ratio in horizontal polarisation was greater than or equal to 5 dB. DSD data were treated as in Raupach and Berne (2017):

5    Parsivel DSDs were truncated to 0.2495 (0.2565) mm to 7 (7.21) mm for HyMeX and Payerne (Iowa) Parsivel data (Raupach and Berne, 2015a); to avoid including overestimated numbers of small drops (Peters et al., 2005), DSDs estimated by the MRR were truncated to 0.6 mm to 5.8 mm (Raupach and Berne, 2017) and MRR data were further subset to records with $R \leq 150$ mm h$^{-1}$ (thus removing 0.2% of records); MRR data for altitudes greater than 2200 m were excluded because not enough points were available at those altitudes; and all DSDs were subset to time steps in which $R > 0.1$ mm h$^{-1}$. In each data set,

10    more than 85% of the DSDs sampled were classified as stratiform type by Raupach and Berne (2017).

     To compare measured versus estimated or retrieved values in this work, we use the median relative bias, the interquartile range (IQR) of relative bias, and the squared Pearson correlation coefficient ($r^2$) between reference and estimated values. If $V_R$ is the reference value and $V_E$ is the estimated value, the relative bias expressed as a percentage of the reference value is defined as $100(V_E - V_R)/V_R$.

15   **5   DSD retrieval from polarimetric radar data**

Raupach and Berne (2017) showed that with reasonably chosen input moments, the double-moment normalised DSD of Lee et al. (2004) can be assumed invariant across spacial displacement in stratiform rain, with a performance loss that is acceptable for practical applications. Results on limited data for non-stratiform rain types suggested that while the double-moment normalised DSD varies more in these cases, the assumption of its invariance may still lead to acceptable performance with input

20    moments that are not both of low or both of high order. Using the assumption of an invariant double-moment normalised DSD model, the DSD can be estimated using polarimetric radar data. Given a known double-moment normalised DSD, the task of DSD reconstruction becomes that of estimating from radar information the values of two DSD moments. In this section we present a new DSD-retrieval method that uses this idea. The aim of the proposed DSD-retrieval technique is to retrieve two DSD moments using only polarimetric radar data.

25    The SCOP-ME method was trained with DSDs simulated using a DSD model and a wide range of DSD parameter values. In contrast, we used empirical DSDs measured by Parsivels to train our method, to avoid any assumption about the shape of the DSD. A trade-off that must be made is that the measured DSDs are truncated. However, previous studies have shown that if the considered range of drop diameters is large enough around the median drop diameter $D_0$ [mm], the effect of truncation on calculated bulk variables is limited (Willis, 1984; Vivekanandan et al., 2004). Willis (1984) concluded that the effect of

30    maximum considered drop size $D_{max}$ on bulk variables is negligible if $D_{max}$ exceeds $2.5D_0$. Using $D_0$ calculated from the recorded (truncated) Parsivel DSDs, this criteria was met for 99.6% of the records. The criteria of Vivekanandan et al. (2004) is that, for there to be less than five percent error on bulk variables, the minimum drop size $D_{min}$ should be less than $D_0/2$ and $D_{max}$ should exceed $4D_0$. This constraint was met by 90.4% of the DSDs (93.5% met this criteria for the upper drop

size limit). Calculated $D_0$ may also be subject to error because of the truncation, but we consider that these calculations give broad confidence in the bulk variables we used to train the method. Further, the truncation on the Parsivel data effects primarily very small drops since large drops are rare, and therefore its influence on the higher-order moments we use is expected to be negligible.

5      The training data set was sampled as 60% of each of the three Parsivel data sets, and contained 182079 measured one-minute DSDs. $Z_H$, $K_{dp}$ and $Z_{DR}$ were calculated for these DSDs for the MXPol stacked PPI incidence angles, temperatures of five, 10, and 15 degrees C, and each of four drop axis ratio functions: those of Andsager et al. (1999), Brandes et al. (2002), Thurai et al. (2007), and that of Beard and Chuang (1987) in the form shown in Kalogiros et al. (2013). Unusual records with $Z_{DR}$ or $K_{dp}$ less than or equal to zero (0.16% of all simulated radar records) were excluded.

**10  5.1  Retrieval of DSD moment six**

Radar reflectivity in linear units, $Z_h$ [mm$^6$ m$^{-3}$], is the sixth moment of the DSD in the Rayleigh scattering regime for spherical drops (Bringi and Chandrasekar, 2001). At X-band frequencies, larger drops enter into the Mie scattering regime and differences appear between $M_6$ and $Z_h$. We use the observation that $Z_h$ departs from $M_6$ for heavier rain, and assume that this departure occurs when $Z_H$ is greater than a threshold value. This threshold was determined through comparison of $M_6$ and 15  $Z_h$ for DSDs, classed by $Z_H$ in classes of width 2 dBZ between 10 dBZ and 40 dBZ, and was set to 28 dBZ. For both smaller and larger reflectivity values, a power law relationship was found using orthogonal least squares fitting in log-log space. The resulting relationship is

$$\widehat{M_6} = \begin{cases} Z_h^{1.01} & \text{if } 10\log_{10}(Z_h) \leq 28 \\ 2.71\, Z_h^{0.86} & \text{if } 10\log_{10}(Z_h) > 28. \end{cases} \tag{13}$$

On the training set, median relative bias between $\widehat{M_6}$ and $M_6$ was 0.1%, the IQR of relative bias was 2.5 percentage points, 20  and the $r^2$ value was 0.98. The fitted relationship is shown on samples of training data in Figure 1. Temperature made only limited difference to the fitted parameters: the pre-factor varied from 2.49 to 2.94 for the larger values of $Z_H$, and the other parameters differed by 0.01 or less from the value found for all temperatures combined.

**5.2  Retrieval of DSD moment three**

Retrieving a second, lower-order DSD moment is more difficult than estimating $M_6$, because radar variables are more closely 25  linked to the higher-order moments of the DSD. Using theoretical relationships as much as possible, we present a method to estimate the third moment of the DSD from polarimetric data. As shown in Equation 9, the reflectivity-weighted mean drop axis ratio, $r_z$, is related to a negative power of the differential reflectivity in linear units. In Kalogiros et al. (2013), the reflectivity-weighted and mass-weighted drop axis ratios were assumed to be the same and differences were dealt with through fitting of qualitative relationships between radar variables. A similar approach is taken here. Since $r_z$ and the mass-weighted

[Figure]

**Figure 1.** A sample of 20,000 points from the training set, showing the relationship between radar reflectivity and DSD moment six in dB scale. The one-to-one line is shown in black; the red dashed line shows the fitted relationship of Equation 13. The $Z_H$ threshold of 28 dBZ is shown with a triangle.

mean drop axis ratio $r_m$ are both weighted mean drop axis ratios, we assume that $r_m$ is also related to differential reflectivity, and estimate $r_m$ using a polynomial fit to $Z_{DR}$, such that

$$\widehat{r_m} = \sum_{i=0}^{5} c_i Z_{DR}^i. \tag{14}$$

With our training data, this polynomial of order five produced low relative bias on retrieval of $M_3$. Recall from Equation 5 that $M_3$ relates to $W$: substituting Equation 5 into Equation 12, and solving for $M_3$, we have

$$M_3 = \frac{\lambda}{0.003\pi C \rho_w} \frac{K_{dp}}{(1 - r_m)}. \tag{15}$$

At X-band (9.4 GHz, $\lambda = 3.189$ cm), assuming that $\rho_w = 1$ g cm$^{-3}$, and replacing $r_m$ with its estimate based on $Z_{DR}$, $M_3$ is predicted by

$$M_3 = \frac{338.4}{\widehat{C}} \frac{K_{dp}}{(1 - \widehat{r_m})}, \tag{16}$$

where $\widehat{C}$ is a single representative value for $C$.

$K_{dp}$ is sensitive to the raindrop axis ratio (e.g. Bringi and Chandrasekar, 2001), so values for $c_i$ and $\widehat{C}$ were found per axis ratio function. The coefficients $c_i$ in Equation 14 were found using least-squares polynomial fitting. In rare cases for large values of $Z_{DR}$ the relationships returned unrealistic values of $r_m$ ($0 \leq r_m$ or $r_m \geq 1$). In these few cases, $\widehat{r_m}$ was set to 0.75. Estimated $\widehat{r_m}$ values were used to find $C$ for each training DSD, and the mean of these values was used as $\widehat{C}$. The results and

**Table 3.** Fitted values of $\widehat{C}$ (Equation 16) and $c_i$ (Equation 14), by drop axis ratio function (Ratio). $M_3$ estimation performance in the training data is shown in terms of median relative bias (RB [%]), IQR of relative bias (IQR [% pts]), and $r^2$. Max $Z_{DR}$ [dB] shows the maximum value of $Z_{DR}$ each relationship can use.

| Ratio | $\widehat{C}$ | $c_0$ | $c_1$ | $c_2$ | $c_3$ | $c_4$ | $c_5$ | RB | IQR | $r^2$ | Max $Z_{DR}$ |
|---|---|---|---|---|---|---|---|---|---|---|---|
| Thurai | 3.419 | 1 | -0.075720 | 0.046043 | -0.019965 | 0.003264 | -0.000164 | 0.8 | 25 | 0.97 | 7.27 |
| Brandes | 3.274 | 1 | -0.080221 | 0.052613 | -0.023125 | 0.004237 | -0.000279 | -0.8 | 22 | 0.97 | 8.12 |
| Andsager | 3.220 | 1 | -0.092664 | 0.074970 | -0.036663 | 0.007466 | -0.000549 | -0.3 | 21 | 0.97 | 7.17 |
| Beard | 3.202 | 1 | -0.088418 | 0.054731 | -0.021392 | 0.003208 | -0.000149 | -0.7 | 22 | 0.97 | 7.47 |

their performance statistics are shown in Table 3. Fitted parameters differed across the three tested temperatures. However, parameters fitted using all training data performed similarly on training data for individual temperatures, with the median relative bias remaining within $\pm 1\%$ of the all-temperatures value, and IQR of relative bias varying by less than one percentage point. The values fitted using combined training data were used.

**5.3 Summary of DSD-retrieval technique**

The proposed DSD retrieval method is summarised as follows: the double-moment normalised DSD $\hat{h}(x)$ with parameters $c$ and $\mu$ is assumed trained from data and known. Then, given $K_{dp}$, $Z_{DR}$ and $Z_h$, (1) DSD moment six is estimated using Equation 13, and (2) DSD moment three is estimated using Equations 14 and 16 and parameters from Table 3. The DSD is then retrieved using Equation 4 with $i = 3$ and $j = 6$.

**6 Comparison to an existing DSD-retrieval method**

The new DSD retrieval method was compared to SCOP-ME (Anagnostou et al., 2009, 2010; Kalogiros et al., 2013). We implemented SCOP-ME using its description in Anagnostou et al. (2013). SCOP-ME was developed for X-band using simulated DSDs and T-matrix simulations of radar variables, and in Anagnostou et al. (2013) it is shown to outperform the algorithms of Anagnostou et al. (2008) and Park et al. (2005a). The DSD model used by SCOP-ME is based on the normalised DSD of Willis (1984) (see also Bringi and Chandrasekar, 2001). Kalogiros et al. (2013) provided an explicit expression for rain rate using polarimetric variables, but since we are interested in the whole DSD, in the following we compare $R$ computed from reconstructed DSDs. The comparison of the two methods is first shown using Parsivel data in which the radar values were simulated using T-matrix codes and were therefore free of radar measurement noise.

Comparisons of the two techniques were made using the Parsivel validation data set composed of 40% of the records from HyMeX, Payerne, and Iowa. For each one-minute DSD record, $Z_h$, $K_{dp}$ and $Z_{DR}$ were calculated using T-matrix codes, for an elevation angle of $4°$ above horizontal, and using each of the four drop axis ratio functions. For the double-moment technique, the generalised gamma model $\hat{h}(x)$ (Equation 4) for $i = 3$ and $j = 6$ was used. $\hat{h}(x)$ was fitted to non-zero median values of $h(x)$ in classes of $x$ with width 0.2, using weighted least squares fitting in log space, with each class weighted by its number

of observations (Raupach and Berne, 2017). The parameters found for the combined Parsivel training data were $c = 0.54$ and $\mu = 3.06$. SCOP-ME and the double-moment method were used to retrieve the DSD concentrations $N(D)$ for $D$ in the class centres of the truncated Parsivel diameter classes. For each technique and axis ratio function, retrieved DSDs were compared to measured DSDs by comparing moments zero to seven, $D_m$ and $R$.

5    Comparisons of relative error distributions by technique are shown in Figure 2. Example scatter plot results are shown for the HyMeX data set and the drop axis ratio model of Beard and Chuang (1987) in Figure 3. The Beard model, which has been shown to match well to observations (Thurai et al., 2009), is shown because it provided the equilibrium drop shapes around which the SCOP-ME training set was simulated (Kalogiros et al., 2013). Full performance results are shown for the HyMeX data set in Table A1, for Payerne in Table A2, and for Iowa in Table A3. The metrics used were median relative bias, 10    IQR of relative bias, $r^2$, and the slope of the linear regression on measured vs. reconstructed points. Differences between the performance metrics for the two techniques were calculated such that a negative difference indicates that the double-moment technique performed better than SCOP-ME. These differences are shown visually in Figure 4, in which red colours show negative differences.

**Table 4.** Average differences between double-moment and SCOP-ME techniques, on Parsivel data, over three regions and four raindrop axis ratios. Negative values show an improvement by the double-moment technique over SCOP-ME.

| Variable | RB | IQR | $r^2$ | Slope |
|---|---|---|---|---|
| $D_m$ | 0.19 | 1.07 | 0.04 | -0.07 |
| $M_0$ | 3.67 | 11.97 | -0.01 | 0.03 |
| $M_1$ | 0.44 | 4.44 | 0.04 | 0.09 |
| $M_2$ | -0.23 | 0.89 | 0.02 | -0.04 |
| $M_3$ | -1.28 | 0.46 | 0.00 | -0.00 |
| $M_4$ | -0.61 | 2.03 | -0.00 | 0.06 |
| $M_5$ | -0.49 | 2.36 | -0.00 | -0.02 |
| $M_6$ | -1.57 | -0.22 | -0.01 | -0.11 |
| $M_7$ | -2.65 | 8.55 | -0.03 | -0.05 |
| $R$ | -0.98 | 2.06 | 0.00 | 0.07 |

In over half of the tested region, axis ratio function, and variable combinations, the double-moment technique produced a 15    better median relative bias than the SCOP-ME technique, with an overall average difference of -0.35 percentage points. IQR of relative bias was usually slightly higher for the double-moment technique, with an average difference of 3.4 percentage points. Correlation coefficients and scatter plot slopes were usually similar for both techniques. The average differences across the three tested regions and four tested raindrop axis ratio functions are shown in Table 4. On average, the double-moment technique produced better median relative bias than SCOP-ME on $R$ and DSD moments two to seven. IQRs were similar on 20    average, with the exception of moments zero, one, and seven for which SCOP-ME produced notably smaller IQRs. As shown in Tables A1, A2, and A3, the results differed across the different drop axis ratio functions and regions. It was often the case that SCOP-ME produced a less biased estimate of DSD moment zero, but in many of these cases the double-moment technique

[Figure]

**Figure 2.** Relative bias distributions for the double-moment and SCOP-ME DSD-retrieval methods, by drop axis ratio function and data set (H stands for HyMeX, P for Payerne, and I for Iowa). Variables are moment order $n$ [mm$^n$ m$^{-3}$], $D_m$ [mm], and $R$ [mm h$^{-1}$]. Bold bars show medians, boxes show IQRs, whiskers show 10th to 90th percentile ranges

produced a better $r^2$. The double-moment technique's performance variations relate to the accuracy of the prediction of DSD moment three from $K_{\mathrm{dp}}$ and $Z_{\mathrm{DR}}$, and to the fit of the generalised gamma function $\hat{h}(x)$. $\hat{h}(x)$ was trained on data from all data sets combined, in order to have the most general model possible. Our experiments showed that performance for low-order moments could be increased in any one region by training the gamma model on data from that region only. This aligns with the conclusions of Raupach and Berne (2017), who noted that while the double-normalised DSD can be assumed invariant for practical purposes, some residual variability remains and results in performance loss that depends on the input moments used. We now move to testing the two techniques on measured radar data, in which noise is a problem that must be dealt with.

[Figure]

**Figure 3.** Density scatter plots of retrieved versus measured moments $M_n$ [$mm^n$ $m^{-3}$], $R$ [$mm\,h^{-1}$], and $D_m$ [mm] for the double-moment method, on the HyMeX data set, using the axis ratio function of Beard and Chuang (1987). One-to-one lines are shown in black. Regression lines for the double-moment method are shown in solid red, and dotted red lines show linear regressions for SCOP-ME, for which the densities are not shown. Values of $D_m$ above 5 mm are extremely rare; less than 0.02% of DSDs in each data set show these values.

**7 Reducing the effects of noise**

Radar data is noisy at light rain rates, particularly for $K_{dp}$ and $Z_{DR}$ (e.g. Bringi et al., 2002; Schneebeli et al., 2014). We propose here a method to deal with this noise for the current application of DSD retrieval. Regressions on $Z_h$ and $\xi_{dr}$ are used

[Figure]

**Figure 4.** Differences in performance between the double-moment technique and SCOP-ME, using radar variables simulated from Parsivel data, by region and drop axis ratio function (differences in Tables A1, A2, and A3). Reds indicate negative differences, where the double-moment technique outperformed SCOP-ME. Variables are moment order $n$ [mm$^n$ m$^{-3}$], $D_m$ [mm], and $R$ [mm h$^{-1}$]. Differences are shown for median relative bias (RB [% pts]), IQR of relative bias (IQR [% pts]), $r^2$ (difference in deviations from unity, multiplied by 100 for display on this scale), and regression slope ($S$, difference in deviations from unity, multiplied by 100).

to determine "expected" values for these variables, which can be used when the measured values are likely to be noisy. We found that $Z_{\mathrm{DR}}$ can be reasonably predicted from $Z_h$ using

$$\hat{Z}_{\mathrm{DR}} \sim \alpha_Z Z_h^{\beta_Z}, \tag{17}$$

and $K_{\mathrm{dp}}$ can be predicted from $Z_h$ and $\xi_{\mathrm{dr}}$ using

$$\quad \hat{K}_{\mathrm{dp}} \sim \alpha_K Z_h^{\beta_{K1}} \xi_{\mathrm{dr}}^{\beta_{K2}} \tag{18}$$

**Table 5.** Fitted coefficients and the performance of the fits on the training data, for Equations 17 and 18, by raindrop axis ratio function (Ratio). Performance is shown in terms of median relative bias (RB [%]) and the IQR [% pts] of relative bias.

| | | | $Z_{DR}$ performance | | | | | $K_{dp}$ performance | |
| Ratio | $\alpha_Z$ | $\beta_Z$ | RB | IQR | $\alpha_K$ | $\beta_{K1}$ | $\beta_{K2}$ | RB | IQR |
|---|---|---|---|---|---|---|---|---|---|
| Thurai | 0.030 | 0.436 | -5 | 64 | 0.00010 | 1.057 | -3.171 | -1 | 19 |
| Brandes | 0.027 | 0.449 | -4 | 70 | 0.00010 | 1.037 | -2.723 | -1 | 13 |
| Andsager | 0.043 | 0.377 | -3 | 57 | 0.00017 | 0.976 | -3.262 | 0 | 16 |
| Beard | 0.048 | 0.384 | -4 | 59 | 0.00017 | 1.015 | -3.365 | -0 | 13 |

with parameters $\alpha_Z$, $\beta_Z$, $\alpha_K$, $\beta_{K1}$ and $\beta_{K2}$. Least-squares fitting in log-log space, using the training data set described in Section 5, was used to find best-fitting parameter values per raindrop axis ratio function. Just as for the retrieval of DSD moment six, assumed air temperature made only a small difference (parameter values fitted to individual temperature data sets differed by less than 4% from those fitted using combined temperatures), whereas different axis ratios produced more diverse parameter values. Resulting parameter values and performance statistics are shown in Table 5.

Threshold values are used to determine when $K_{dp}$ and $Z_{DR}$ may be noisy. A threshold value on $Z_H$ selects values of $Z_H$ for which $K_{dp}$ and $Z_{DR}$ showed large variation around their expected values in the three radar data sets used here. Threshold values on $Z_{DR}$ and $K_{dp}$ are those of Bringi et al. (2002). To reduce the effects of noise, then, if $Z_H < 37$ dBZ or $Z_{DR} < 0.2$ dB, measured $Z_{DR}$ is replaced by the the expected value $\hat{Z}_{DR}$ and $\xi_{dr}$ is replaced by $10^{(\hat{Z}_{DR}/10)}$. Likewise, if $Z_H < 37$ dBZ or $K_{dp} < 0.3°$ km$^{-1}$, $K_{dp}$ is replaced by $\hat{K}_{dp}$ (calculated with $\hat{\xi}_{dr}$ if $\xi_{dr}$ was replaced). This treatment method allows radar data with negative or zero $K_{dp}$ or $Z_{DR}$ to be used. The treatment improved DSD-retrieval performance for both the double-moment and SCOP-ME techniques. For example, when retrieved DSDs were matched to measured MRR data (Section 8.1), the median relative bias was reduced by an average (across variables) of ∼6 percentage points for SCOP-ME and by ∼16 percentage points for the double-moment technique, while average IQRs were reduced more; for example on the comparison with MRR data the IQRs were reduced by ∼81 (69) percentage points for the SCOP-ME (double-moment) method. When retrieved DSDs were compared to Parsivel data (Section 8.2), the noise in the radar data contributed to errors to such an extent that for both techniques the proposed treatment reduced the IQR and at times the median of relative bias by hundreds of percentage points for some variables. We note that because most values of $Z_H$ recorded in the PPIs analysed here were lower than 37 dBZ, the noise correction affected the majority of radar records.

**8 Comparisons using radar data**

The DSD-retrieval techniques were applied to PPI radar data from the three locations. The double-moment technique was run on noise-corrected data. SCOP-ME was run on uncorrected PPI data (subset to $K_{dp} > 0$ and $Z_{DR} > 0$) and noise-corrected data. We used the elevation angles of the stacked PPIs for HyMeX, 5° for Payerne, and 3° for Iowa. Measured radar variables $Z_H$, $K_{dp}$ and $Z_{DR}$ were recovered for volumes corresponding to instrument locations. DSD retrieval was performed using these

values, and the resulting DSDs compared to those that were measured by other instruments. All comparisons using PPI data involved a difference in measurement volume – a change-of-support problem that we expect will introduce error spread (e.g. Raupach and Berne, 2016). There were, at times, significant vertical distances between the radar volume and the ground-based Parsivels used in these comparisons (see Table 2). These factors and uncertainty in the noise correction technique combine to

5    create greater uncertainty in the comparisons of the two techniques made using real data than in those made using simulated radar variables from disdrometer data.

Because the axis ratio of Thurai et al. (2007) produced good results using the double-moment technique on the Parsivel data, the double-moment technique was used with parameters for this axis ratio function. Note that the assumption of axis ratio function affects only parameters of the double-moment technique, because the radar data used in this section are measured, not

10    simulated, and the SCOP-ME technique is used as presented in Anagnostou et al. (2013). In the HyMeX campaign, the lowest available PPI elevation angle (4°) was used to compare results to Parsivels, but there was also an MRR at Pradel Grainage which retrieved estimates of the DSD aloft. MRR-derived DSDs were compared at eight different altitudes using the MXPol stacked PPIs (except 20° elevation) above the HyMeX instrument network. We first address the comparisons with MRR for HyMeX, then move to the comparisons with the Parsivel networks in all three regions.

15    **8.1    Comparisons to MRR DSD estimates aloft**

MXPol volume centre altitudes were projected onto MRR altitude classes for comparison. The double-moment DSD-retrieval algorithm was used with generalised gamma model $\hat{h}$ parameters (Equation 4) for MRR data and $i = 3$ and $j = 6$. These parameters were found using the same fitting technique as for Parsivel data (Section 6), but differ since instrumental differences produce different forms of $h(x)$ (Raupach and Berne, 2017). The parameters were set to $c = 0.4$ and $\mu = 32.25$ (the value of

20    $\mu$ was reduced from 32.28 to stay within the computational limits of the software used). The large value of $\mu$ is due to the large numbers of small drops returned by the DSD-retrieval algorithm used by the MRR (Raupach and Berne, 2017), and is compensated somewhat by the small value of $c$. The reconstructed DSDs were found for classes of drop diameter from 0.65 to 5.75 mm with a class width of 0.1 mm, so that the reconstructed truncation matched that of the MRR data. PPI values from eight 100 m altitude classes between about 900 and 2100 m above sea level were compared to MRR estimates of the DSD

25    aloft. Two output pairings are shown here: the first in which both techniques used noise-corrected data, and the second in which the SCOP-ME technique used raw data and the double-moment technique used the same raw data set corrected for noise. This second pairing was made to ensure that the performance of SCOP-ME was not compromised by the noise-correction technique.

Results of comparisons between MRR- and PPI-derived DSDs are shown for three example altitudes in Figure 5. There was good agreement between the recorded radar reflectivity recorded by both instruments, with a median relative bias of −3%, an

30    IQR on relative bias of 16 percentage points, and a value of $r^2$ of 0.63. The improvement in SCOP-ME performance made by the noise correction is clear. When both techniques used noise-corrected input, both overestimated DSD moment orders zero to four and underestimated orders six and seven. Rain rate was recovered with a median relative bias of 2% (IQR 94 % pts) by the double-moment technique and 17% (IQR 106 % pts) by SCOP-ME. The double-moment technique showed lower median relative bias than SCOP-ME on moments one to four, seven, and $R$, and smaller IQRs on moment two to six, $D_m$, and

$R$. Similar to some of the Parsivel results, the double-moment technique overestimated moments zero and one of the DSD. $r^2$ values were low for both techniques (the maximum was 0.31, by SCOP-ME for $D_m$), but the double-moment technique had the same or a slightly higher value of $r^2$ in the majority of cases. High best-fit slopes were observed for both techniques for moments five, six, and seven, and show the effect of a few outlier points in these cases. Performance differences between the two techniques using noise-corrected data are shown in Table A4. Overall, the double-moment technique for DSD-retrieval out-performed SCOP-ME for the retrieval of DSD moments above order zero and rain rate measured aloft by the MRR.

[Figure]

**Figure 5.** Distributions of relative bias on DSD moments zero to seven, comparing DSDs retrieved using PPI data to those measured by the MRR at Pradel Grainage. The results are classed by altitude for a selection of three altitudes across the compared range. Two comparisons are shown: in comparison A, SCOP-ME used raw PPI data and the double-moment technique used the noise-corrected version of the same data set. In B, both techniques used noise-corrected data sets. Symbols as for Figure 2.

**8.2 Comparisons to DSDs measured by Parsivels**

DSDs retrieved from polarimetric radar data were also compared to those recorded by ground-based Parsivels in the three regions we studied. Unlike in previous sections where training and validation divisions of the Parsivel data were used, here we compared DSDs derived using independent radar data to all available matching Parsivel records. The DSDs were retrieved in truncated Parsivel drop diameter classes, using the Parsivel generalised gamma model parameters quoted in Section 6. In the Payerne and Iowa data sets, the noise-correction routine was required in order to retrieve realistic DSDs; the results shown here are thus for the SCOP-ME and double-moment techniques both run on noise-corrected PPI data. Figure 6 shows distributions of DSD-retrieval relative error for each region.

[Figure]

**Figure 6.** Distributions of relative bias on DSD moments zero to seven, comparing DSDs retrieved using noise-corrected PPI data, and those measured by Parsivel networks. Symbols as for Figure 2.

[Figure]

**Figure 7.** Differences in performance between the double-moment technique and SCOP-ME using noise-corrected radar data, for MRR and for Parsivels by region (differences in Table A4). Variables and performance statistics as for Figure 4. Red indicates that the double-moment technique outperformed SCOP-ME. Grey indicates an $r^2$ difference greater than 50 on this scale; these slopes were affected by outliers.

The double-moment technique produced smaller ranges of relative bias than SCOP-ME for moments five, six, and seven. For moment orders zero and one, the double-moment technique produced better median relative bias than SCOP-ME in the HyMeX and Iowa data sets, but worse in Payerne. Where the double-moment technique produced better median relative bias, the average improvement was of four percentage points, while in cases where SCOP-ME performed better, the average improvement was five percentage points. Values of $r^2$ and scatter plot slope were similar between the two techniques, with the majority of cases showing differences of less than 0.05 for both variables. Differences in performance between the two techniques are shown in Figure 7 and Table A4.

The performance of the double-moment technique is reliant on how accurately two DSD moments can be extracted from radar data, and in turn on how accurate the radar data are. Both retrieval techniques appear to be similarly affected by radar inaccuracies such as bias in $Z_H$, and experiments with different reflectivity bias corrections (not shown here) showed similar patterns of results. In Parsivel comparisons, the proposed DSD-retrieval technique was applied using a single double-moment normalised DSD model in all three tested regions, without significant performance loss between regions. This supports previous findings (Raupach and Berne, 2017) that for practical use with real radar data in primarily stratiform rain, the double-moment normalised DSD may be considered invariant in regions at similar latitudes.

**9 Conclusions**

Given the assumption of an invariant normalised DSD, and an estimate of that function, the DSD can be predicted using only two of its moments using the double-moment normalisation method of Lee et al. (2004). Two DSD moments are available from polarimetric radar data. At X-band, radar reflectivity can be used to accurately predict the sixth moment of the DSD, and moment three can be retrieved relatively accurately using $K_{dp}$ and $Z_{DR}$. We showed that by estimating these two DSD moments from radar data, the DSD for a radar volume can be predicted using the double-moment formulation. Tests on disdrometer data from three networks in different climatic regions showed that DSD-retrieval using this new technique produced similar or slightly better performance than the SCOP-ME DSD-retrieval technique of Kalogiros et al. (2013). The proposed method is also more flexible, because there is no prescribed functional form for the double-moment normalised DSD, and even a non-parametric $\hat{h}(x)$ could be used. Nor is there a prescribed method of DSD moment extraction, which means that the moments used could be tailored to the intended purpose.

A new method for treatment of radar data with possibly noisy values of $K_{dp}$ and $Z_{DR}$ was proposed. The method is based on predicting the expected values of these variables from radar reflectivity, and considerably improved the performance of both the DSD-retrieval techniques. DSDs were predicted from polarimetric variables in noise-corrected PPI scans measured by X-band radars in each of the three regions. Comparisons of the retrieved DSDs to MRR data for DSDs aloft in the HyMeX region in France, and of radar-retrieved DSDs to disdrometer data from the three regions, showed reasonable agreement but large error spread for both methods. This study provides a proof-of-concept for DSD-retrieval using noise-corrected radar data, the double-moment normalisation method of (Lee et al., 2004), and a generalised gamma model for the normalised DSD. Performance improvements may be possible through future work, that should test the approach using different instruments and data sets, address more precise prediction of low-order DSD moments from polarimetric radar data, and investigate different models and fitting methods for the double-moment normalised DSD.

*Acknowledgements.* For deploying and maintaining the instruments used, we thank, for HyMeX: J. Jaffrain, M. Schleiss, J. Grazioli, D. Wolfensberger, A. Studzinski (EPFL LTE), B. Boudevillain, G. Molinié, S. Gérard (Laboratoire d'étude des Transferts en Hydrologie et Environnement (LTHE), Grenoble University), Y. Pointin and J. Van Baelen (Laboratoire de Météorologie Physique (LaMP), Université Blaise Pascal de Clermont-Ferrand). For Payerne: J. Jaffrain and Meteosuisse. We thank J. Grazioli for processing the MRR and radar data sets.

HyMeX data were obtained from the HyMeX program, sponsored by grants MISTRALS/HyMeX, ANR-2011-BS56-027 FLOODSCALE project, OHMCV & EPFL-LTE. We thank the Swiss National Science Foundation for financial support under grant 2000021_140669. We thank two anonymous reviewers for their constructive reviews.

**Table A1.** Comparison of double-moment method to SCOP-ME results on all Parsivel data in the HyMeX data set by axis ratio function (Ratio). RB [%] is median relative bias, IQR [% pts] is interquartile range of relative bias [% points], $r^2$ is squared correlation coefficient. S is the slope on measured vs. reconstructed regression. Difference is difference in absolute values for RB and IQR, and difference in deviation from unity for $r^2$ and slope. A negative difference shows that the double-moment method improved on SCOP-ME's performance.

| Ratio | Var | Double-moment | | | | SCOP-ME | | | | Difference | | | |
|---|---|---|---|---|---|---|---|---|---|---|---|---|---|
| | | RB | IQR | $r^2$ | S | RB | IQR | $r^2$ | S | RB | IQR | $r^2$ | S |
| Ands. | $D_m$ | −1 | 10 | 0.85 | 0.99 | −2 | 9 | 0.88 | 0.83 | −1 | 1 | 0.04 | −0.17 |
| | $M_0$ | 20 | 80 | 0.70 | 0.97 | 13 | 70 | 0.66 | 0.98 | 7 | 10 | −0.04 | 0.01 |
| | $M_1$ | 10 | 52 | 0.79 | 0.97 | 12 | 50 | 0.80 | 1.09 | −1 | 2 | 0.02 | −0.06 |
| | $M_2$ | 6 | 34 | 0.89 | 1.03 | 10 | 34 | 0.91 | 1.13 | −4 | −0 | 0.02 | −0.10 |
| | $M_3$ | 4 | 21 | 0.97 | 1.09 | 8 | 21 | 0.97 | 1.07 | −4 | 0 | 0.00 | 0.02 |
| | $M_4$ | 3 | 12 | 0.99 | 1.09 | 6 | 11 | 0.99 | 0.99 | −3 | 1 | 0.00 | 0.08 |
| | $M_5$ | 1 | 6 | 0.99 | 1.03 | 3 | 4 | 0.99 | 0.93 | −2 | 2 | 0.00 | −0.04 |
| | $M_6$ | 0 | 3 | 0.98 | 0.94 | 2 | 3 | 0.98 | 0.89 | −2 | 1 | −0.01 | −0.06 |
| | $M_7$ | −1 | 14 | 0.98 | 0.83 | 1 | 3 | 0.94 | 0.83 | 0 | 11 | −0.04 | 0.00 |
| | $R$ | 3 | 14 | 0.99 | 1.11 | 6 | 12 | 0.98 | 1.02 | −3 | 2 | 0.00 | 0.09 |
| Thur. | $D_m$ | −1 | 13 | 0.83 | 0.99 | −0 | 12 | 0.87 | 0.78 | 1 | 1 | 0.04 | −0.21 |
| | $M_0$ | 14 | 94 | 0.64 | 0.91 | 2 | 74 | 0.55 | 0.89 | 13 | 20 | −0.09 | −0.03 |
| | $M_1$ | 7 | 64 | 0.75 | 0.94 | 1 | 55 | 0.73 | 1.05 | 6 | 9 | −0.02 | 0.02 |
| | $M_2$ | 4 | 42 | 0.88 | 1.02 | 1 | 39 | 0.88 | 1.15 | 3 | 3 | 0.00 | −0.13 |
| | $M_3$ | 4 | 26 | 0.96 | 1.09 | 1 | 26 | 0.96 | 1.14 | 3 | 1 | 0.00 | −0.05 |
| | $M_4$ | 3 | 14 | 0.99 | 1.09 | 1 | 14 | 0.99 | 1.06 | 2 | 0 | 0.00 | 0.03 |
| | $M_5$ | 1 | 7 | 0.99 | 1.03 | 1 | 5 | 0.99 | 0.98 | 0 | 2 | 0.00 | 0.02 |
| | $M_6$ | 0 | 2 | 0.99 | 0.94 | 2 | 2 | 0.98 | 0.92 | −2 | 0 | −0.01 | −0.02 |
| | $M_7$ | −1 | 12 | 0.98 | 0.81 | 4 | 5 | 0.94 | 0.84 | −3 | 7 | −0.04 | 0.03 |
| | $R$ | 3 | 16 | 0.99 | 1.12 | 1 | 16 | 0.99 | 1.09 | 2 | 0 | 0.00 | 0.02 |
| Bran. | $D_m$ | −0 | 11 | 0.83 | 0.99 | 1 | 10 | 0.87 | 0.80 | −0 | 0 | 0.04 | −0.18 |
| | $M_0$ | 14 | 79 | 0.66 | 0.90 | −3 | 66 | 0.58 | 0.86 | 11 | 13 | −0.08 | −0.04 |
| | $M_1$ | 5 | 53 | 0.75 | 0.92 | −3 | 48 | 0.75 | 1.00 | 3 | 5 | −0.01 | 0.08 |
| | $M_2$ | 2 | 35 | 0.87 | 1.02 | −2 | 34 | 0.88 | 1.08 | −1 | 1 | 0.01 | −0.07 |
| | $M_3$ | 0 | 23 | 0.96 | 1.10 | −2 | 22 | 0.96 | 1.07 | −1 | 1 | 0.00 | 0.03 |
| | $M_4$ | 0 | 15 | 0.99 | 1.11 | −1 | 12 | 0.99 | 1.00 | −1 | 3 | 0.00 | 0.10 |
| | $M_5$ | −1 | 7 | 0.99 | 1.06 | −0 | 4 | 0.99 | 0.94 | 0 | 3 | 0.00 | −0.01 |
| | $M_6$ | 0 | 3 | 0.98 | 0.98 | 2 | 3 | 0.98 | 0.89 | −2 | −0 | −0.01 | −0.08 |
| | $M_7$ | 1 | 15 | 0.98 | 0.87 | 5 | 6 | 0.94 | 0.84 | −5 | 9 | −0.04 | −0.03 |
| | $R$ | −0 | 17 | 0.98 | 1.12 | −1 | 13 | 0.98 | 1.02 | −1 | 4 | 0.00 | 0.10 |
| Beard | $D_m$ | −1 | 10 | 0.85 | 0.99 | −1 | 10 | 0.88 | 0.80 | −0 | 1 | 0.04 | −0.19 |
| | $M_0$ | 15 | 78 | 0.69 | 0.93 | 6 | 69 | 0.61 | 0.94 | 9 | 9 | −0.08 | 0.02 |
| | $M_1$ | 6 | 52 | 0.78 | 0.94 | 5 | 50 | 0.77 | 1.08 | 1 | 2 | −0.01 | −0.02 |
| | $M_2$ | 3 | 34 | 0.89 | 1.03 | 4 | 34 | 0.90 | 1.16 | −1 | −0 | 0.01 | −0.13 |
| | $M_3$ | 2 | 22 | 0.97 | 1.10 | 4 | 21 | 0.97 | 1.13 | −2 | 1 | 0.00 | −0.02 |
| | $M_4$ | 2 | 13 | 0.99 | 1.11 | 3 | 11 | 0.99 | 1.05 | −1 | 3 | 0.00 | 0.07 |
| | $M_5$ | 1 | 7 | 0.99 | 1.06 | 2 | 4 | 1.00 | 0.98 | −1 | 4 | 0.01 | 0.03 |
| | $M_6$ | 1 | 3 | 0.98 | 0.96 | 2 | 3 | 0.98 | 0.93 | −1 | −0 | 0.00 | −0.03 |
| | $M_7$ | 0 | 14 | 0.98 | 0.83 | 3 | 4 | 0.95 | 0.86 | −2 | 10 | −0.03 | 0.03 |
| | $R$ | 2 | 15 | 0.99 | 1.13 | 3 | 12 | 0.99 | 1.08 | −1 | 3 | 0.00 | 0.05 |

**Table A2.** Comparison of double-moment method to SCOP-ME results on all Parsivel data in the Payerne data set by axis ratio function (Ratio). Columns are as for Table A1.

| Ratio | Var | Double-moment | | | | SCOP-ME | | | | Difference | | | |
|---|---|---|---|---|---|---|---|---|---|---|---|---|---|
| | | RB | IQR | $r^2$ | S | RB | IQR | $r^2$ | S | RB | IQR | $r^2$ | S |
| Ands. | $D_m$ | 1 | 8 | 0.85 | 1.15 | −0 | 7 | 0.88 | 0.92 | 1 | 1 | 0.04 | 0.08 |
| | $M_0$ | 10 | 63 | 0.60 | 0.88 | −1 | 45 | 0.70 | 0.89 | 9 | 17 | 0.09 | 0.01 |
| | $M_1$ | 2 | 40 | 0.74 | 0.89 | 1 | 35 | 0.80 | 0.96 | 1 | 5 | 0.06 | 0.07 |
| | $M_2$ | −0 | 25 | 0.87 | 0.94 | 3 | 25 | 0.89 | 0.99 | −3 | −0 | 0.02 | 0.05 |
| | $M_3$ | −0 | 14 | 0.96 | 1.00 | 4 | 16 | 0.95 | 0.98 | −4 | −2 | −0.01 | −0.01 |
| | $M_4$ | 1 | 8 | 0.99 | 1.05 | 4 | 8 | 0.98 | 0.94 | −3 | −0 | −0.01 | −0.01 |
| | $M_5$ | 1 | 4 | 0.99 | 1.05 | 3 | 3 | 0.98 | 0.84 | −2 | 1 | −0.01 | −0.11 |
| | $M_6$ | 0 | 2 | 0.98 | 0.93 | 2 | 2 | 0.96 | 0.71 | −2 | −0 | −0.02 | −0.22 |
| | $M_7$ | 0 | 9 | 0.96 | 0.76 | 1 | 2 | 0.95 | 0.59 | −1 | 7 | −0.01 | −0.17 |
| | $R$ | 0 | 9 | 0.99 | 1.04 | 4 | 9 | 0.97 | 0.97 | −4 | −0 | −0.01 | 0.01 |
| Thur. | $D_m$ | 1 | 10 | 0.83 | 1.18 | 2 | 9 | 0.87 | 0.90 | −1 | 2 | 0.04 | 0.08 |
| | $M_0$ | 7 | 73 | 0.56 | 0.85 | −11 | 46 | 0.66 | 0.82 | −4 | 27 | 0.11 | −0.03 |
| | $M_1$ | 0 | 48 | 0.70 | 0.86 | −8 | 36 | 0.78 | 0.90 | −8 | 12 | 0.08 | 0.04 |
| | $M_2$ | −1 | 31 | 0.84 | 0.91 | −6 | 27 | 0.88 | 0.95 | −4 | 4 | 0.04 | 0.04 |
| | $M_3$ | −0 | 19 | 0.95 | 0.97 | −3 | 18 | 0.95 | 0.97 | −3 | 0 | 0.00 | 0.00 |
| | $M_4$ | 1 | 10 | 0.99 | 1.03 | −1 | 10 | 0.98 | 0.96 | −0 | −0 | −0.01 | −0.02 |
| | $M_5$ | 1 | 4 | 0.99 | 1.03 | 0 | 4 | 0.98 | 0.87 | 1 | 0 | −0.01 | −0.10 |
| | $M_6$ | 0 | 2 | 0.98 | 0.92 | 2 | 2 | 0.96 | 0.73 | −2 | −0 | −0.02 | −0.19 |
| | $M_7$ | −0 | 7 | 0.96 | 0.75 | 4 | 4 | 0.95 | 0.61 | −4 | 3 | −0.01 | −0.15 |
| | $R$ | 0 | 11 | 0.98 | 1.02 | −1 | 12 | 0.98 | 0.98 | −1 | −1 | 0.00 | 0.00 |
| Bran. | $D_m$ | 2 | 10 | 0.83 | 1.15 | 3 | 9 | 0.87 | 0.90 | −1 | 1 | 0.04 | 0.05 |
| | $M_0$ | 3 | 65 | 0.57 | 0.83 | −14 | 44 | 0.66 | 0.80 | −12 | 21 | 0.09 | −0.03 |
| | $M_1$ | −3 | 44 | 0.72 | 0.86 | −12 | 35 | 0.78 | 0.87 | −8 | 10 | 0.06 | 0.02 |
| | $M_2$ | −4 | 29 | 0.85 | 0.92 | −9 | 26 | 0.88 | 0.93 | −4 | 3 | 0.03 | 0.01 |
| | $M_3$ | −3 | 18 | 0.95 | 1.00 | −5 | 18 | 0.95 | 0.96 | −3 | 0 | 0.00 | −0.04 |
| | $M_4$ | −1 | 10 | 0.99 | 1.08 | −3 | 10 | 0.98 | 0.94 | −2 | 0 | 0.00 | 0.02 |
| | $M_5$ | −0 | 5 | 0.99 | 1.10 | −0 | 4 | 0.98 | 0.87 | −0 | 1 | 0.00 | −0.03 |
| | $M_6$ | 0 | 2 | 0.98 | 0.99 | 3 | 2 | 0.97 | 0.74 | −2 | −0 | −0.01 | −0.25 |
| | $M_7$ | 1 | 9 | 0.96 | 0.82 | 6 | 4 | 0.96 | 0.63 | −4 | 5 | −0.01 | −0.19 |
| | $R$ | −2 | 11 | 0.98 | 1.06 | −3 | 11 | 0.98 | 0.96 | −1 | 0 | −0.01 | 0.02 |
| Beard | $D_m$ | 1 | 9 | 0.84 | 1.14 | 1 | 8 | 0.88 | 0.89 | 0 | 1 | 0.04 | 0.03 |
| | $M_0$ | 6 | 61 | 0.60 | 0.86 | −7 | 45 | 0.68 | 0.87 | −1 | 16 | 0.08 | 0.01 |
| | $M_1$ | −1 | 40 | 0.74 | 0.89 | −5 | 34 | 0.80 | 0.95 | −4 | 6 | 0.06 | 0.06 |
| | $M_2$ | −3 | 25 | 0.87 | 0.95 | −2 | 25 | 0.89 | 1.00 | 0 | 1 | 0.03 | 0.05 |
| | $M_3$ | −2 | 16 | 0.96 | 1.02 | −0 | 16 | 0.96 | 1.02 | 1 | −0 | 0.00 | 0.00 |
| | $M_4$ | −0 | 10 | 0.99 | 1.07 | 1 | 9 | 0.99 | 0.99 | −1 | 1 | −0.01 | 0.06 |
| | $M_5$ | 1 | 5 | 0.99 | 1.07 | 2 | 3 | 0.98 | 0.90 | −1 | 2 | 0.00 | −0.03 |
| | $M_6$ | 1 | 2 | 0.98 | 0.94 | 2 | 2 | 0.97 | 0.75 | −1 | −0 | −0.01 | −0.19 |
| | $M_7$ | 1 | 9 | 0.96 | 0.76 | 3 | 3 | 0.96 | 0.63 | −2 | 6 | 0.00 | −0.13 |
| | $R$ | −1 | 11 | 0.99 | 1.07 | 1 | 10 | 0.98 | 1.01 | −0 | 1 | 0.00 | 0.05 |

**Table A3.** Comparison of double-moment method to SCOP-ME results on all Parsivel data in the Iowa data set by axis ratio function (Ratio). Columns are as for Table A1.

| Ratio | Var | Double-moment | | | | SCOP-ME | | | | Difference | | | |
|---|---|---|---|---|---|---|---|---|---|---|---|---|---|
| | | RB | IQR | $r^2$ | S | RB | IQR | $r^2$ | S | RB | IQR | $r^2$ | S |
| Ands. | $D_m$ | 2 | 13 | 0.86 | 1.07 | 0 | 12 | 0.91 | 0.85 | 2 | 1 | 0.04 | −0.08 |
| | $M_0$ | −11 | 72 | 0.53 | 0.61 | −4 | 73 | 0.48 | 0.75 | 6 | −1 | −0.04 | 0.14 |
| | $M_1$ | −9 | 52 | 0.70 | 0.74 | −2 | 54 | 0.75 | 1.03 | 7 | −2 | 0.05 | 0.23 |
| | $M_2$ | −6 | 36 | 0.89 | 0.92 | 0 | 37 | 0.92 | 1.11 | 6 | −1 | 0.03 | −0.03 |
| | $M_3$ | −2 | 25 | 0.97 | 1.09 | 2 | 23 | 0.98 | 1.04 | 0 | 2 | 0.01 | 0.04 |
| | $M_4$ | 1 | 17 | 0.99 | 1.14 | 2 | 12 | 1.00 | 0.97 | −1 | 5 | 0.01 | 0.11 |
| | $M_5$ | 0 | 8 | 0.99 | 1.08 | 1 | 5 | 0.99 | 0.91 | −1 | 4 | 0.00 | −0.02 |
| | $M_6$ | 1 | 4 | 0.99 | 0.93 | 2 | 4 | 0.98 | 0.85 | −1 | 0 | −0.01 | −0.08 |
| | $M_7$ | 0 | 19 | 0.98 | 0.77 | 2 | 5 | 0.95 | 0.79 | −2 | 14 | −0.03 | 0.02 |
| | $R$ | 1 | 19 | 0.99 | 1.16 | 2 | 14 | 0.99 | 1.00 | −1 | 5 | 0.01 | 0.16 |
| Thur. | $D_m$ | 2 | 15 | 0.86 | 1.10 | 1 | 13 | 0.90 | 0.83 | 1 | 2 | 0.04 | −0.07 |
| | $M_0$ | −11 | 85 | 0.46 | 0.59 | −9 | 78 | 0.39 | 0.69 | 2 | 7 | −0.06 | 0.10 |
| | $M_1$ | −9 | 63 | 0.66 | 0.71 | −6 | 59 | 0.70 | 1.03 | 3 | 4 | 0.05 | 0.26 |
| | $M_2$ | −4 | 44 | 0.88 | 0.88 | −3 | 42 | 0.91 | 1.16 | 1 | 2 | 0.03 | −0.04 |
| | $M_3$ | 0 | 27 | 0.98 | 1.04 | −1 | 27 | 0.98 | 1.10 | −1 | 0 | 0.01 | −0.06 |
| | $M_4$ | 3 | 15 | 0.99 | 1.11 | 0 | 14 | 1.00 | 1.01 | 3 | 1 | 0.00 | 0.09 |
| | $M_5$ | 2 | 7 | 0.99 | 1.05 | 0 | 5 | 0.99 | 0.93 | 1 | 2 | 0.00 | −0.02 |
| | $M_6$ | 1 | 3 | 0.99 | 0.92 | 2 | 3 | 0.97 | 0.86 | −1 | −0 | −0.01 | −0.06 |
| | $M_7$ | −1 | 14 | 0.98 | 0.77 | 3 | 6 | 0.95 | 0.80 | −3 | 8 | −0.03 | 0.02 |
| | $R$ | 3 | 17 | 0.99 | 1.11 | 0 | 16 | 0.99 | 1.05 | 2 | 1 | 0.01 | 0.06 |
| Bran. | $D_m$ | 2 | 14 | 0.86 | 1.10 | 1 | 12 | 0.90 | 0.86 | 0 | 1 | 0.04 | −0.04 |
| | $M_0$ | −9 | 77 | 0.61 | 0.77 | −10 | 71 | 0.56 | 0.82 | −1 | 6 | −0.05 | 0.05 |
| | $M_1$ | −8 | 56 | 0.71 | 0.76 | −8 | 52 | 0.76 | 1.01 | 1 | 3 | 0.05 | 0.22 |
| | $M_2$ | −5 | 38 | 0.89 | 0.90 | −5 | 37 | 0.92 | 1.08 | 0 | 1 | 0.03 | 0.02 |
| | $M_3$ | −2 | 25 | 0.97 | 1.08 | −3 | 24 | 0.98 | 1.03 | −1 | 1 | 0.01 | 0.05 |
| | $M_4$ | 1 | 17 | 0.99 | 1.15 | −2 | 12 | 1.00 | 0.96 | −1 | 4 | 0.01 | 0.11 |
| | $M_5$ | 0 | 8 | 0.99 | 1.10 | −1 | 5 | 0.99 | 0.90 | −1 | 3 | 0.00 | 0.01 |
| | $M_6$ | 1 | 3 | 0.99 | 0.97 | 2 | 4 | 0.97 | 0.85 | −1 | −1 | −0.01 | −0.12 |
| | $M_7$ | 1 | 17 | 0.98 | 0.82 | 4 | 8 | 0.95 | 0.80 | −3 | 9 | −0.04 | −0.02 |
| | $R$ | −0 | 18 | 0.98 | 1.16 | −2 | 14 | 0.99 | 0.99 | −2 | 4 | 0.01 | 0.15 |
| Beard | $D_m$ | 2 | 13 | 0.86 | 1.07 | 0 | 12 | 0.91 | 0.83 | 2 | 1 | 0.04 | −0.10 |
| | $M_0$ | −9 | 76 | 0.52 | 0.62 | −5 | 78 | 0.45 | 0.77 | 4 | −3 | −0.07 | 0.14 |
| | $M_1$ | −8 | 55 | 0.70 | 0.75 | −2 | 58 | 0.73 | 1.08 | 6 | −3 | 0.04 | 0.17 |
| | $M_2$ | −4 | 38 | 0.89 | 0.94 | 0 | 40 | 0.92 | 1.18 | 4 | −2 | 0.02 | −0.12 |
| | $M_3$ | −0 | 26 | 0.98 | 1.09 | 2 | 25 | 0.98 | 1.11 | −2 | 1 | 0.00 | −0.01 |
| | $M_4$ | 3 | 18 | 0.99 | 1.14 | 2 | 13 | 1.00 | 1.01 | 0 | 6 | 0.00 | 0.13 |
| | $M_5$ | 1 | 9 | 0.99 | 1.08 | 2 | 4 | 0.99 | 0.93 | −0 | 5 | 0.00 | 0.00 |
| | $M_6$ | 1 | 3 | 0.99 | 0.94 | 2 | 3 | 0.97 | 0.86 | −1 | −0 | −0.02 | −0.08 |
| | $M_7$ | 0 | 18 | 0.98 | 0.78 | 3 | 5 | 0.95 | 0.79 | −2 | 13 | −0.03 | 0.01 |
| | $R$ | 2 | 20 | 0.99 | 1.16 | 2 | 15 | 0.99 | 1.06 | −0 | 6 | 0.00 | 0.11 |

**Table A4.** Differences in performance by variable and region, for DSDs retrieved from noise-corrected PPI data using the double-moment technique and SCOP-ME, compared to the MRR at Pradel Grainage (MRR) and Parsivels by region (HyMeX, Payerne, and Iowa). Metrics and differences are defined as for Table A1. An exception is $Z$, which refers to $Z_H$ measured by the radar, not reconstructed through DSD-retrieval (hence it is the same for both techniques).

| | | Double-moment | | | | SCOP-ME | | | | Difference | | | |
|---|---|---|---|---|---|---|---|---|---|---|---|---|---|
| | Variable | RB | IQR | $r^2$ | S | RB | IQR | $r^2$ | S | RB | IQR | $r^2$ | S |
| MRR | $D_m$ | $-7$ | 26 | 0.28 | 0.42 | $-6$ | 26 | 0.31 | 0.43 | 1 | $-1$ | 0.03 | 0.01 |
| | $M_0$ | 26 | 181 | 0.00 | 0.01 | 17 | 160 | 0.00 | 0.01 | 9 | 21 | 0.00 | 0.00 |
| | $M_1$ | 21 | 156 | 0.01 | 0.02 | 24 | 149 | 0.01 | 0.02 | $-3$ | 7 | 0.00 | 0.00 |
| | $M_2$ | 19 | 133 | 0.01 | 0.05 | 27 | 134 | 0.01 | 0.04 | $-7$ | $-1$ | 0.00 | 0.00 |
| | $M_3$ | 11 | 107 | 0.02 | 0.12 | 22 | 113 | 0.01 | 0.11 | $-11$ | $-5$ | $-0.01$ | 0.00 |
| | $M_4$ | $-1$ | 88 | 0.04 | 0.44 | 12 | 99 | 0.03 | 0.41 | $-11$ | $-11$ | $-0.01$ | $-0.03$ |
| | $M_5$ | $-8$ | 74 | 0.13 | 1.79 | 3 | 83 | 0.11 | 1.56 | 6 | $-9$ | $-0.02$ | 0.23 |
| | $M_6$ | $-15$ | 78 | 0.26 | 3.61 | $-12$ | 82 | 0.18 | 2.97 | 3 | $-4$ | $-0.08$ | 0.64 |
| | $M_7$ | $-19$ | 91 | 0.29 | 4.07 | $-26$ | 88 | 0.18 | 3.21 | $-7$ | 3 | $-0.11$ | 0.86 |
| | $R$ | 2 | 94 | 0.03 | 0.27 | 17 | 106 | 0.03 | 0.26 | $-15$ | $-12$ | 0.00 | $-0.01$ |
| | $Z$ | $-3$ | 16 | 0.63 | 0.89 | $-3$ | 16 | 0.63 | 0.89 | 0 | 0 | 0.00 | 0.00 |
| HyMeX | $D_m$ | $-6$ | 26 | 0.29 | 0.50 | $-5$ | 25 | 0.31 | 0.43 | 1 | 1 | 0.02 | $-0.07$ |
| | $M_0$ | $-3$ | 150 | 0.02 | 0.28 | $-10$ | 120 | 0.04 | 0.26 | $-7$ | 30 | 0.02 | $-0.02$ |
| | $M_1$ | $-10$ | 122 | 0.04 | 0.36 | $-13$ | 110 | 0.07 | 0.35 | $-3$ | 12 | 0.03 | $-0.01$ |
| | $M_2$ | $-14$ | 104 | 0.11 | 0.46 | $-16$ | 100 | 0.13 | 0.45 | $-2$ | 4 | 0.02 | $-0.01$ |
| | $M_3$ | $-20$ | 95 | 0.21 | 0.49 | $-21$ | 93 | 0.21 | 0.48 | $-2$ | 1 | 0.00 | $-0.01$ |
| | $M_4$ | $-26$ | 94 | 0.22 | 0.41 | $-26$ | 93 | 0.24 | 0.41 | $-1$ | 1 | 0.02 | 0.00 |
| | $M_5$ | $-32$ | 96 | 0.13 | 0.30 | $-32$ | 96 | 0.19 | 0.31 | 0 | $-0$ | 0.06 | 0.01 |
| | $M_6$ | $-39$ | 100 | 0.05 | 0.21 | $-37$ | 105 | 0.11 | 0.22 | 2 | $-6$ | 0.06 | 0.01 |
| | $M_7$ | $-46$ | 106 | 0.02 | 0.16 | $-41$ | 118 | 0.06 | 0.17 | 5 | $-12$ | 0.04 | 0.01 |
| | $R$ | $-21$ | 97 | 0.24 | 0.47 | $-22$ | 96 | 0.23 | 0.46 | $-1$ | 2 | $-0.01$ | $-0.01$ |
| | $Z$ | $-9$ | 26 | 0.50 | 0.68 | $-9$ | 26 | 0.50 | 0.68 | 0 | 0 | 0.00 | 0.00 |
| Payerne | $D_m$ | $-7$ | 22 | 0.21 | 0.34 | $-5$ | 22 | 0.21 | 0.30 | 2 | $-0$ | 0.00 | $-0.03$ |
| | $M_0$ | 31 | 171 | 0.07 | 0.50 | $-4$ | 124 | 0.11 | 0.34 | 27 | 47 | 0.04 | $-0.16$ |
| | $M_1$ | 17 | 138 | 0.08 | 0.43 | $-1$ | 121 | 0.11 | 0.36 | 16 | 17 | 0.03 | $-0.08$ |
| | $M_2$ | 6 | 119 | 0.12 | 0.43 | $-1$ | 114 | 0.14 | 0.40 | 5 | 5 | 0.02 | $-0.03$ |
| | $M_3$ | $-4$ | 106 | 0.19 | 0.46 | $-5$ | 105 | 0.20 | 0.46 | $-2$ | 2 | 0.01 | $-0.01$ |
| | $M_4$ | $-13$ | 99 | 0.29 | 0.48 | $-12$ | 99 | 0.27 | 0.50 | 1 | 0 | $-0.02$ | 0.01 |
| | $M_5$ | $-22$ | 103 | 0.29 | 0.37 | $-21$ | 104 | 0.26 | 0.41 | 1 | $-1$ | $-0.03$ | 0.04 |
| | $M_6$ | $-31$ | 107 | 0.12 | 0.12 | $-29$ | 110 | 0.11 | 0.15 | 2 | $-3$ | $-0.01$ | 0.02 |
| | $M_7$ | $-42$ | 109 | 0.03 | 0.02 | $-40$ | 116 | 0.02 | 0.03 | 2 | $-8$ | $-0.01$ | 0.01 |
| | $R$ | $-7$ | 108 | 0.26 | 0.51 | $-6$ | 107 | 0.25 | 0.51 | 1 | 1 | $-0.01$ | 0.01 |
| | $Z$ | $-8$ | 28 | 0.42 | 0.57 | $-8$ | 28 | 0.42 | 0.57 | 0 | 0 | 0.00 | 0.00 |
| Iowa | $D_m$ | $-10$ | 28 | 0.33 | 0.39 | $-11$ | 26 | 0.37 | 0.34 | $-1$ | 1 | 0.04 | $-0.04$ |
| | $M_0$ | 8 | 111 | 0.08 | 0.72 | 26 | 129 | 0.09 | 0.56 | $-18$ | $-18$ | 0.01 | $-0.16$ |
| | $M_1$ | 7 | 100 | 0.14 | 0.79 | 19 | 118 | 0.14 | 0.67 | $-12$ | $-18$ | 0.00 | $-0.13$ |
| | $M_2$ | 6 | 109 | 0.22 | 0.75 | 11 | 114 | 0.22 | 0.68 | $-5$ | $-5$ | 0.00 | $-0.08$ |
| | $M_3$ | 4 | 106 | 0.31 | 0.59 | 4 | 105 | 0.30 | 0.56 | $-0$ | 1 | $-0.01$ | $-0.02$ |
| | $M_4$ | $-5$ | 107 | 0.33 | 0.37 | $-6$ | 105 | 0.32 | 0.39 | $-1$ | 2 | $-0.01$ | 0.02 |
| | $M_5$ | $-17$ | 109 | 0.25 | 0.19 | $-15$ | 110 | 0.28 | 0.23 | 2 | $-1$ | 0.03 | 0.04 |
| | $M_6$ | $-32$ | 116 | 0.13 | 0.08 | $-25$ | 123 | 0.18 | 0.12 | 7 | $-7$ | 0.05 | 0.03 |
| | $M_7$ | $-44$ | 114 | 0.04 | 0.03 | $-35$ | 135 | 0.09 | 0.06 | 10 | $-21$ | 0.05 | 0.02 |
| | $R$ | 4 | 113 | 0.33 | 0.49 | 2 | 109 | 0.32 | 0.50 | 3 | 4 | $-0.01$ | 0.01 |
| | $Z$ | $-5$ | 25 | 0.54 | 0.62 | $-5$ | 25 | 0.54 | 0.62 | 0 | 0 | 0.00 | 0.00 |